

# The soccer season: performance variations and evolutionary trends

Joao Renato Silva

Center for Research, Education, Innovation, and Intervention in Sport (CIFI2D), Faculty of Sport, University of Porto, Portugal

## ABSTRACT

The physiological demands of soccer challenge the entire spectrum of the response capacity of the biological systems and fitness requirements of the players. In this review we examined variations and evolutionary trends in body composition, neuromuscular and endurance-related parameters, as well as in game-related physical parameters of professional players. Further, we explore aspects relevant for training monitoring and we reference how different training stimulus and situational variables (*e.g.*, competition exposure) affect the physiological and performance parameters of players. Generally, improvements of small magnitude in non- (non-CMJ) and countermovement-based jumps (CMJ$_{Based}$) and in the sprint acceleration (ACC$_{Phase}$) and maximal velocity phase (MV$_{Phase}$) are observed from start of preparation phase (PPS) to beginning of competition phase (BCP). A greater magnitude of increases is observed in physiological and endurance performance measures within this period; moderate magnitude in sub-maximal intensity exercise (velocity at fixed blood lactate concentrations; V$_{2–4mmol/l}$) and large magnitude in VO$_{2max}$, maximal aerobic speed (MAS) and intense intermittent exercise performance (IE). In the middle of competition phase (MCP), small (CMJ$_{Based}$ and ACC$_{Phase}$), moderate (non-CMJ; MV$_{Phase}$; VO$_{2max}$; sub-maximal exercise) and large (MAS and IE) improvements were observed compared to PPS. In the end of competition period (ECP), CMJ$_{Based}$ and MV$_{Phase}$ improve to a small extent with non-CMJ, and ACC$_{Phase}$, VO$_{2max}$, MAS, sub-maximal intensity exercise and IE revealing moderate increments compared to PPS. Although less investigated, there are generally observed alterations of trivial magnitude in neuromuscular and endurance-related parameters between in-season assessments; only substantial alterations are examined for IE and sub-maximal exercise performance (decrease and increase of small magnitude, respectively) from BCP to MCP and in VO$_{2max}$ and IE (decrements of small magnitude) from MCP to ECP. Match performance may vary during the season. Although, the variability between studies is clear for TD, VHSR and sprint, all the studies observed substantial increments in HSR between MCP and ECP. Finally, studies examining evolutionary trends by means of exercise and competition performance measures suggests of a heightened importance of neuromuscular factors. In conclusion, during the preseason players "recover" body composition profile and neuromuscular and endurance competitive capacity. Within in-season, and more robustly towards ECP, alterations in neuromuscular performance seem to be force-velocity dependent, and in some cases, physiological determinants and endurance performance may be compromised when considering other in-season moments. Importantly, there is a substantial variability in team responses that can be observed during in-season. Consequently,

Corresponding author
Joao Renato Silva,
jm_silv@hotmail.com

this informs on the need to both provide a regular training stimulus and adequate monitorization throughout the season.

# INTRODUCTION

In soccer, during both training practices and matches, players perform a wide range of activities (*e.g.*, sprints) that demand players be able to sustain and produce forceful contractions (*Stolen et al., 2005*). Moreover, there is evidence (*e.g.*, global positioning systems) suggesting that the mechanical and metabolic loads imposed during training and games is even higher that previously suspected (*Barnes et al., 2014*; *Bush et al., 2014*; *Konefal et al., 2019b*; *Osgnach et al., 2010*; *Varley & Aughey, 2013*). The repeated bouts of intermittent soccer-specific activities of an aerobic and/or anaerobic nature impose acute and chronic strains on various physiological systems (*e.g.*, musculoskeletal, nervous, and metabolic) that may lead to declines and impairments in performance (*e.g.*, reductions in strength/power-based parameters), biological functions (*e.g.*, hormonal milieu, biochemical responses) and perceptual responses (*e.g.*, muscle soreness) in different players (*Bangsbo, Mohr & Krustrup, 2006*; *Kraemer et al., 2004*; *Reilly, Drust & Clarke, 2008*; *Reinke et al., 2009*; *Silva et al., 2013a*; *Silva et al., 2014*; *Smith et al., 2018*).

Notwithstanding the evidence that there are physiological characteristics that favor the capacity of playing a specific field position in soccer (*Altmann et al., 2020*; *Carling & Orhant, 2010*; *Konefal et al., 2019b*), the game demands sufficient skills such that substantial deviations from this profile remain compatible with a high standard of performance (*Shephard, 1999*). Nevertheless, a positive body composition (*e.g.*, low adiposity) and proficient neuromuscular (*e.g.*, strength and power) and endurance-related (*e.g.*, high-intensity intermittent exercise) qualities provide a competitive advantage, as they are associated with improved fatigue resistance during the game (*Bangsbo, Iaia & Krustrup, 2008*; *Silva et al., 2018*) faster post-match recovery (*Hader et al., 2019*; *Owen et al., 2015*; *Tofari, Kemp & Cormack, 2017*) and injury prevention (*Al Attar et al., 2017*; *Malone et al., 2019*; *Malone et al., 2016*; *Zouita et al., 2016*). As so, players perform intense training programs to potentiate these fitness determinants to cope with the acute and chronic demands of a high-level soccer season cycle (*Brocherie et al., 2014*; *Chmura et al., 2019*; *Eliakim et al., 2018*; *Malone et al., 2018*; *Silva et al., 2014*). Therefore, to prevent performance decline and to ensure that training programs are effective, elite clubs should have as a required organizational practice the implementation of a training monitoring system (*e.g.*, performance tests, records of daily exercise intensity) and effective strategies to aid in player recovery (*Silva & Rebelo, 2019*).

## Rationale for the review

Player's physical performance is one of the relevant performance domains and so, understanding the dynamic nature of adaptations throughout the season is of relevance for

the population of soccer players. However, despite that there are important reviews concerning physiological characteristics of soccer players (*Shephard, 1999*; *Stolen et al., 2005*; *Svensson & Drust, 2005*), soccer biomechanics (*Lees et al., 2010*; *Lees & Nolan, 1998*), determinants of players' performance (*Bangsbo, Iaia & Krustrup, 2007*; *Bangsbo, Mohr & Krustrup, 2006*; *Reilly, Drust & Clarke, 2008*), specific training-induced effects (*Hill-Haas et al., 2011*; *Hoff, 2005*; *Hoff & Helgerud, 2004*; *Iaia, Rampinini & Bangsbo, 2009*; *Silva, Nassis & Rebelo, 2015*), and development of soccer fatigue and kinetics of recovery (*Hader et al., 2019*; *Mohr, Krustrup & Bangsbo, 2005*; *Nedelec et al., 2012*; *Nedelec et al., 2013*; *Reilly, Drust & Clarke, 2008*; *Reilly & Ekblom, 2005*; *Silva et al., 2018*), an understanding of seasonal adaptations and evolutionary trends on players' physical fitness is still required.

### Intended audience and organization

Understanding the different variables that affect the dynamic nature of adaptations during the season may allow coaches, medical departments, and researchers to improve training periodization and monitoring. We target this review for students (*e.g.*, exercise physiology and strength and conditioning), researchers, and all practitioners (coaches and medical department related staff) to whom the knowledge about the physiological and functional characteristics of the players is a matter of undeniable interest and understanding the different variables that affect the dynamic nature of adaptations occurring within the training season may allow informed decisions on training periodization and monitoring.

In this review we examined adaptations in body composition (body mass, body fat and lean body mass) and different neuromuscular qualities of professional soccer players (force production, jump, sprint and change of direction abilities). Subsequently, we analyzed seasonal alterations in endurance-related physiological and performance parameters as well as in competition measures of professional soccer players. Moreover, we will reference how different training stimulus and situational variables (*e.g.*, competition exposure) affect the physiological and performance parameters of highly trained players. Further, we will explore aspects relevant for training monitoring of professional players.

## SURVEY METHODOLOGY

### Literature search strategy

For the search for relevant scientific literature, a review was performed using the PubMed and SportDiscus databases multiple times until June 2022. Additionally, Google Scholar and bibliographic searches of relevant articles were also completed. The description of seasonal trends comprises papers from January 2000 to April 2022 (Tables 1 and 2).

The search strategy included the following search terms and Boolean operators using the term "soccer" AND "seasonal alterations", OR "performance analysis", OR "competition", OR "physiology", OR "body composition", OR "strength training", OR "neuromuscular performance", OR "fatigue", OR "field tests", OR "intermittent endurance", OR "muscular power", OR "jump ability", OR "sprint ability", OR "agility", OR "change of direction", OR "training period", OR "detraining", OR "off season", OR "in season", OR "preseason" OR "competition period".

**Table 1  Studies included in the quantitative description of seasonal variations in physical fitness.**

| Study | Sample | N (Age) | PPS | BCP | MCP | ECP |
|---|---|---|---|---|---|---|
| (Aziz, Tan & Teh, 2005) | Elite Singapore | 41 (25.7) | X | X | X | X |
| (Bonuccelli et al., 2012) | Elite Italy | 10 (26.7) | X | X | X | X |
| (Boullosa et al., 2013) | Elite Spain | 12 (24) | X | X |  | X |
| (Bradley et al., 2011) | Elite Denmark | 10 (adult) | X | X | X | X |
| (Bunc, Hráský & Skalská, 2015) | Elite England | 15 (U19) | X | X | X | X |
|  | Elite Czech Republic | 45 (21.9) | X |  | X | X |
| (Campos-Vazquez et al., 2016) | Professional Spain | 12 (27.7) | X | X | X | X |
| (Casajús, 2001) | Elite Spain | 15 (25.8) | X |  | X | X |
| (Castagna et al., 2011) | Elite Italy | 14 (25) | X | X | X |  |
| (Castagna et al., 2013) | Elite Italy | 18 (28.6) | X | X |  |  |
| (Clark et al., 2008) | Elite England | 10–22 (25) | X |  | X | X |
| (Clemente et al., 2021) | Professional Portugal | 25 (28.1) | X | X |  |  |
| (D'Ascenzi et al., 2013) | Professional Italy | 23 (26.6) | X | X | X | X |
| (Devlin et al., 2017) | Elite Australia | 18 (25.5) | X | X | X | X |
| (Dunbar, 2002) | Professional England | 11 (NS-adults) | X | X | X | X |
| (Edwards, Macfadyen & Clark, 2003) | Professional England | 12 (26.2) | X |  |  | X |
| (Eliakim et al., 2018) | Professional Israel | 31 (NS-adults) | X | X | X |  |
| (Eniseler et al., 2012) | Elite Turkey | 14 (25.8) | X |  |  |  |
| (Fessi et al., 2016) | Professional Qatarl | 17 (23.7) | X | X | X | X |
| (Haritodinis et al., 2004) | Elite Greece | 12 (25) | X | X | X | X |
| (Iaia et al., 2009b) | Professional Denmark | 12 (22.4) | X | X | X |  |
| (Iga et al., 2014) | Professional England | 35 (20.4) | X | X | X | X |
| (Kalapotharakos, Ziogas & Tokmakidis, 2011) | Elite Greece | 12 (25) | X | X | X |  |
| (Koundourakis et al., 2014) | Professional Greece | 22–23 (23.8–25.5) | X | X |  | X |
| (Krustrup et al., 2003) | Elite Denmark | 10 (adult) | X | X | X | X |
| (Krustrup et al., 2006) | 1st & 2nd League Denmark | 15–20 (adult) | X | X |  | X |
| (Lago-Peñas et al., 2013) | Professional Spain | 42 (25) | X | X | X |  |
| (Link & de Lorenzo, 2016) | Professional Germany | 428 (adult) | X | X |  | X |
| (Los Arcos et al., 2015) | Professional Spain | 14 (20.6) | X | X |  |  |
| (Malliou et al., 2003) | Professional Greece | 19 (27.2) | X | X |  |  |
| (Manzi et al., 2013) | Elite Italy | 18 (28.4) | X | X |  |  |
| (Meckel et al., 2018) | Professional Israel | 18 (22–32) | X | X | X |  |
| (Metaxas et al., 2006) | Elite Greece | 10–12 (18.1–18.2) | X | X | X | X |
| (Michalczyk et al., 2008) | Professional Poland | 19 (26.1) | X | X |  |  |
| (Mohr, Krustrup & Bangsbo, 2003) | Professional Denmark | 10 (26.4) | X | X | X | X |
| (Mohr, Krustrup & Bangsbo, 2002) | Elite Denmark | 11 (24.0) | X | X | X |  |
| (Morgans et al., 2014) | Professional England | 6 (25.7) |  | X | X | X |
| (Ostojic, 2003) | Elite Serbia | 30 (23.5) | X | X | X | X |
| (Ostojic et al., 2009) | Elite Serbia | 12 (25.8) | X | X |  |  |
| (Owen et al., 2018) | Elite European | 22 (24) | X | X |  |  |
| (Padron-Cabo et al., 2018) | Professional Spain | 519 (adult) |  | X | X | X |
| (Papadakis, Patras & Georgouli, 2015) | Professional Greece | 10 (23.6) | X | X | X | X |
| (Rampinini et al., 2007b) | Professional Italy | 20 (26.4) | X | X | X | X |
| (Reinke et al., 2009) | Professional Germany | 10 (20–36) | X | X | X | X |
| (Requena et al., 2017) | Professional Spain | 19 (26.2) | X |  | X | X |
| (Silva et al., 2013b) | Professional Portugal | 13 (25.7) | X | X | X | X |
| (Silva et al., 2011) | Professional Portugal | 18 (25.7) | X | X | X | X |
| (Suda et al., 2012) | Professional Japan | 21 (24.7) | X | X | X | X |
| (Zoppi et al., 2006) | Professional Brazil | 10 (18.2) | X | X |  |  |

**Note:**
N, sample size; PPS, prior pre-season; BCP, beginning competition phase; MCP, middle competition phase; ECP, end competition phase.

**Table 2 Studies included in the quantitative description of seasonal variations by outcome.**

| Study | Body Composition | Strength | Jump ability | | Sprint ability | | | Endurance | | | | GPP |
|---|---|---|---|---|---|---|---|---|---|---|---|---|
| | | | Non-CMJ | $CMJ_{Based}$ | $ACC_{Phase}$ | $MV_{Phase}$ | COD | $VO_{2max}$ | MAS | SM | IE | |
| (*Aziz, Tan & Teh, 2005*) | BM; BF | | $SJ_{WAS}$ | | 5–20-m | | | E | | | | |
| (*Bonuccelli et al., 2012*) | BF; LBM | | | | | | | | | | | |
| (*Boullosa et al., 2013*) | | | | | | | | | Gacon Test | | YYIR1 | |
| (*Bradley et al., 2011*) | | | | | | | | | | | YYIE2 | |
| (*Bunc, Hráský & Skalská, 2015*) | BM; BF; LBM | | | | | | | D | Vpeak | | | |
| (*Campos-Vazquez et al., 2016*) | | | | | | | | | | | 30-15 | |
| (*Casajús, 2001*) | BM; BF; LBM | | SJ; $SJ_{WAS}$ | CMJ; $CMJ_{15s}$ | | | | D | | $VT_{Speed}$; $VT_{HR}$; $VT_{VO2}$ | | |
| (*Castagna et al., 2011*) | | | | | | | | | | $V_{@2mmol/l}$ | | |
| (*Castagna et al., 2013*) | | | | | | | | D | | $V_{@2-4mmol/l}$ | YYIR1 | |
| (*Clark et al., 2008*) | | | | CMJ; $CMJ_{20s}$ | | | | D | | $AT_{\%VO2max}$ | | |
| (*Clemente et al., 2021*) | BM; BF | | | | | | | | | | | |
| (*D'Ascenzi et al., 2013*) | BM; BF; LBM | | | | | | | | | | | |
| (*Devlin et al., 2017*) | BF; LBM | | | | | | | | | | | |
| (*Dunbar, 2002*) | | | | | | | | | | $V_{2-3mmol/l}$ | | |
| (*Edwards, Macfadyen & Clark, 2003*) | BM | | | CMJ | | | | D | | $VT_{VO2}$; $LT_{VO2}$ | | |
| (*Eliakim et al., 2018*) | | KE; KF | | | | | | D | | | | |
| (*Eniseler et al., 2012*) | | | | | | | | | | | | |
| (*Fessi et al., 2016*) | BM; BF | | | CMJ; $CMJ_{WAS}$ | 10-m | 30-m | | D | Vam-Eval | | | |
| (*Haritodinis et al., 2004*) | | | | | | | | | | | | |
| (*Iaia et al., 2009b*) | | | | | | | | D | | | YYIR2 | |
| (*Iga et al., 2014*) | BF | | | | | | | | | | | |
| (*Kalapotharakos, Ziogas & Tokmakidis, 2011*) | BM; BF | | | | | | | D | $vVO_{2max}$ | $\%VO_{2max}$ & $HR_{max}$ & $V_{@4mmol/l}$ | | |
| (*Koundourakis et al., 2014*) | BM; BF | | SJ | CMJ | 10–20-m | | | D | | | | |
| (*Krustrup et al., 2003*) | | | | | | | | | | | YYIR1 | |
| (*Krustrup et al., 2006*) | | | | | | | | | | | YYIR2 | |
| (*Lago-Peñas et al., 2013*) | BM; BF | | SJ | CMJ; $CMJ_{WAS}$ | | | | D | Vam-Eval | | | |
| (*Link & de Lorenzo, 2016*) | | | | | | | | | | | | TD |
| (*Los Arcos et al., 2015*) | BM; BF | | | CMJ; $CMJ_{WAS}$ | 5–15-m | | | | | $V_{@3mmol/l}$ | | |

(Continued)

| Study | Body Composition | Strength | Jump ability Non-CMJ | Jump ability CMJ_Based | Sprint ability ACC_Phase | Sprint ability MV_Phase | Sprint ability COD | Endurance VO2max | Endurance MAS | Endurance SM | Endurance IE | GPP |
|---|---|---|---|---|---|---|---|---|---|---|---|---|
| (Malliou et al., 2003) | | KE | SJ | CMJ | | | | | | | | |
| (Manzi et al., 2013) | | | | | | | | D | | $V_{@4mmol/l}$; $VT_{VO2}$ | YYIR1 | |
| (Meckel et al., 2018) | BM; BF | | | CMJ | | | 4 × 10-m | D | | $VT_{Speed}$ | | |
| (Metaxas et al., 2006) | BM; BF; LBM | | | | | | | D | | | | |
| (Michalczyk et al., 2008) | BM | | | | | | | D | | | | |
| (Mohr, Krustrup & Bangsbo, 2003) | | | | | | | | D | | | | TD; HSR |
| (Mohr, Krustrup & Bangsbo, 2002) | | | | | | | | | | $HR_{10-18\ km/h}$ | | |
| (Morgans et al., 2014) | | | | | | 50-m | | | | | | TD; VHSR; Sprint |
| (Ostojic, 2003) | BM; BF; LBM | | | | | | | | | | | |
| (Ostojic et al., 2009) | BM; BF | | | CMJ | | | | | | | | |
| (Owen et al., 2018) | BF; LBM | | | | | | | | | | | |
| (Padron-Cabo et al., 2018) | BF | | | CMJ | | | | | | | | TD; VHSR; Sprint |
| (Papadakis, Patras & Georgouli, 2015) | | | | | | | | | | $V_{@2-4mmol/l}$ | | |
| (Rampinini et al., 2007b) | | | | | | | | | | | | TD; HSR VHSR |
| (Reinke et al., 2009) | BM; BF | | | | | | | | | | | |
| (Requena et al., 2017) | BM; BF; LBM | | | CMJ | 15-m | | | | Vam-Eval | | | |
| (Silva et al., 2013b) | | | | | | | | | | | | TD; HSR; Sprint |
| (Silva et al., 2011) | BM; BF | KE; KF | | CMJ | 5-m | 30-m | T-test | | | | YYIE2 | |
| (Suda et al., 2012) | BM; BF; LBM | | | | | | | | | | | |
| (Zoppi et al., 2006) | | | | | | 30-m | | | | $LT_{Speed}$ | | |

**Note:**

N, sample size; BM, body mass; BF, body fat; LBM, lean body mass; KE, knee extensors in isokinetic mode; Non-CMJ, non-countermovement jump; CMJ_Based, jumps involving a countermovement jump; ACC_Phase, sprint acceleration phase; MV_Phase, maximal velocity phase; COD, change of direction ability; VO2max, maximal oxygen consumption; MAS, maximal aerobic speed; SM, submaximal intensity exercise; IE, intense intermittent exercise; GPP, game physical parameters; CMJ, countermovement jump; CMJ_WAS, countermovement jump with arm swing; SJ, squat jump; SJ_WAS, squat jump with arm swing; E, estimated; D, direct measurement; YYIR1, yo-yo intermittent recovery test level1; YYIR2, yo-yo intermittent recovery test level 2; YYIE2, yo-yo endurance intermittent test level 2; TD, total distance; HSR, high speed running distance; VHSR, very-high speed running distance; GT, Gancon Test; VT_Speed/HR/VO2, Speed/Heart rate/oxygen consumption at ventilatory threshold; V_@2-4mmol/l, speed at a blood lactate concentration of 2, 3 and 4 mmol/l; LT_Speed, speed at lactate threshold; LT_VO2, oxygen consumption at lactate threshold; HR_10-14-18 km/h, heart rate at speed of 10 14 and 18 km/h; %VO2max@4mmol, percentage of VO2max at a blood lactate concentration of 4 mmol/l; HR_max@4mmol, percentage of maximal heart rate at a blood lactate concentration of 4 mmol/l; AT_%VO2max, percentage of VO2max at the anaerobic threshold.

**Analysis and interpretation of the results**

Studies were included if they: (i) investigated adults (>19 years) soccer players described has professional or elite player (ii) measured at least two season time points; specifically, the preparation period (PPS), beginning of the competitive period (BCP), middle (MCP) or end of competition period (ECP).

The mean and standard deviation for each measurement was extracted. In the case the necessary statistics were represented in figures and graphs their value was extrapolated using a specific software for the purpose (*webPlotDigitizer;* https://automeris.io/WebPlotDigitizer/). To evaluate the magnitude of the effects, percent change was calculated for each dependent variable for each study using the procedures defined elsewhere (*Silva et al., 2018*). Using the procedures defined in *Schmitz et al. (2018)* we compute a global mean (by time-point and variation between moments) based on the reported means of the individual studie for each outcome. We apply this procedure assuming that the players from each research within the same group belong to the same population and that their test results were extract from the same normal distribution (*Schmitz et al., 2018*). Each global mean was computed as weighted mean of the individual reported mean, with weights built by the number of subjects per investigation (*Schmitz et al., 2018*). Effect size (ES) were computed to present standardized values on the outcome variables (*Cohen, 1998*). The different ES within individual studies were calculated with Cohen's d, by dividing the raw ES (difference in means) by the pooled standard deviations, as proposed (*Cohen, 1998*). To account for possible overestimation of the true population ESs were corrected accounting for the magnitude of the sample size of each study (*Lakens, 2013*). Therefore, a correction factor was calculated as proposed by *Hedges & Olkin (1985)*. Threshold values for *g* were defined as trivial (<0.2), small (0.2–0.6), moderate (0.6–1.2), large (1.2–2.0) and very large (>2.0) (*Cohen, 1998*).

## ANTHROPOMETRIC AND NEUROMUSCULAR ADAPTATIONS: WHY THE RELEVANCE?

Players body composition analysis is becoming increasingly widespread in professional football and is considered important for help players reach optimal performance potential (*Mills, De Ste Croix & Cooper, 2017*). As example, excessive BF may act as "dead weight" placing an unnecessary "load" and stress on players every time they "compete" against gravity and opponents for conquer a positional advantage during the game. Additionally, improved/increases in "lean body mass" (muscle mass) may favor the execution of the high impulsive actions (*e.g.*, sprint) that are essential from a performance and recovery standpoint (*e.g.*, greater fatigue resistance and decrease muscle damage) (*Malone et al., 2016*; *Owen et al., 2015*; *Silva, 2019*; *Silva & Rebelo, 2019*).

The analysis of the players' activity during games and trainings, along with the physiological, neuromuscular, and perceptual responses to training and competition demands, highlights the important role of neuromuscular function for successful soccer performance (*Nedelec et al., 2012*; *Silva et al., 2018*). The high-impulsive efforts, such as sprints, jumps, acceleration/deceleration, and duels require maximal neuromuscular efforts (*Cometti et al., 2001*). These efforts have the goal of maximize the impulse produced

(*Winter et al., 2016*) as this determines the decisive decision-making situations in professional soccer (*e.g.*, speed) (*Faude, Koch & Meyer, 2012*; *Martinez-Hernandez, Quinn & Jones, 2022*). Consequently, the impulse produced during these muscle actions of both concentric, isometric, and eccentric nature, with more relevance to the latter impose significant stress on the neuromuscular and physiological systems (*Dellal et al., 2010*; *Gaudino et al., 2013*; *Hader et al., 2014*; *Hader et al., 2019*). In effect, a massive mechanical and metabolic load is imposed on players not only during the maximal intensities' phases of the game but also every time acceleration occurs, even when speeds are low (*Osgnach et al., 2010*). These speed and direction of movement changes performed during games place stress on the involved musculature from a metabolic viewpoint, thereby affecting energy usage and resulting in a higher physiologic impact than habitual forward movements (*Buchheit et al., 2010*; *Dellal et al., 2010*). From a mechanical standpoint, an increased eccentric load is associated with exercise-induced muscle damage (*Byrne, Twist & Eston, 2004*), contributing to more rapid development of fatigue (*e.g.*, transient, and residual fatigue; peripheral or central) and consequently increasing the odds of injury.

Other evidence of the relevance of neuromuscular function for actual soccer has been suggested by reports that $VO_{2max}$ values among professional players are not improving over time (*Tonnessen et al., 2013*), and contrasting findings concerning sprinting velocity have been observed, *e.g.*, small but positive inter-seasonal development (*Haugen, Tonnessen & Seiler, 2013*). These facts lead to the suggestion that neuromuscular and anaerobic-related parameters (*e.g.*, sprinting ability) are assuming a greater preponderance in modern soccer than other, more typical endurance parameters (*e.g.*, $VO_{2max}$). Interestingly, there are also indications for greater dominance of neuromuscular factors during game (*Barnes et al., 2014*; *Pons et al., 2021*).

Although not universally confirmed (*Metaxas et al., 2009*), some reports suggest that superior neuromuscular function can be observed in soccer players of a higher standard, which includes greater strength (*Cometti et al., 2001*; *Dauty & Potiron Josse, 2004*), short distance sprint speed (*Cometti et al., 2001*; *Dauty & Potiron Josse, 2004*; *Haugen, Tonnessen & Seiler, 2013*), agility or COD (*Mujika et al., 2008*; *Power, Dunbar & Treasure, 2005*; *Reilly et al., 2000*) and anaerobic endurance (*Power, Dunbar & Treasure, 2005*). In addition, these greater neuromuscular capabilities are not only suggested by the higher ability to perform powerful contractions during isokinetic force production tasks but also during and throughout repetitive stretch-shortening cycle activities (SSC) (*Impellizzeri et al., 2008*; *Mujika et al., 2008*; *Rampinini et al., 2009b*). Given these factors, the recent observation that power and speed abilities are determinants in defining result outcomes is not surprising (*Faude, Koch & Meyer, 2012*; *Martinez-Hernandez, Quinn & Jones, 2022*) and should be considered when monitoring training plans. Moreover, neuromuscular and anaerobic-related qualities of professional players (*e.g.*, sprint capacity, power production) have been associated with higher and improved soccer-specific running capacity and are reflected by the following: (i) the ability to perform high-intensity intermittent endurance exercise tests (*Ingebrigtsen et al., 2013a*; *Wells et al., 2014*); (ii) maximal speed and time to exhaustion of the players during a maximal anaerobic running test being strongly associated with YYIR2 performance (*Wells et al., 2014*); (iii) increments in the former
neuromuscular and anaerobic qualities being associated with improvements in YYIR2 (*Wells et al., 2014*); (iv) high performance in certain game-related physical parameters (*Altmann et al., 2018*) as well as lower fatigue development during the match (*Silva et al., 2013b*) and during the post-match recovery period (*Tofari, Kemp & Cormack, 2017, 2020*); and (v) strength may acts as a moderator of injury occurrence (*Al Attar et al., 2017*; *de Hoyo et al., 2015a*). Along this line of reasoning, recent reports have revealed that professional players with higher chronic competition exposure may show higher performance in muscle power actions (*Morgans, Di Michele & Drust, 2017*; *Silva et al., 2011*; *Sporis et al., 2011*). These facts may also suggest that seasonal alterations in neuromuscular performance may be influenced by competition time – match exposure represents a considerable and important "training" stimulus for improving muscle-power-based actions (*Morgans, Di Michele & Drust, 2017*).

## VARIATIONS IN PHYSIOLOGICAL DETERMINANTS AND NEUROMUSCULAR PERFORMANCE

### Body composition

Studies investigating seasonal changes in anthropometric variables, such as body mass (BM, $n = 507$) (*Aziz, Tan & Teh, 2005*; *Bunc, Hráský & Skalská, 2015*; *Casajus, 2001*; *Clemente et al., 2021*; *D'Ascenzi et al., 2013*; *Edwards, Macfadyen & Clark, 2003*; *Fessi et al., 2016*; *Kalapotharakos, Ziogas & Tokmakidis, 2011*; *Koundourakis et al., 2014*; *Lago-Peñas et al., 2013*; *Meckel et al., 2018*; *Metaxas et al., 2006*; *Michalczyk et al., 2008*; *Ostojic, 2003*; *Ostojic et al., 2009*; *Reinke et al., 2009*; *Silva et al., 2011*; *Suda et al., 2012*), body fat (BF, $n = 579$) (*Aziz, Tan & Teh, 2005*; *Bonuccelli et al., 2012*; *Bunc, Hráský & Skalská, 2015*; *Casajus, 2001*; *Clemente et al., 2021*; *D'Ascenzi et al., 2013*; *Devlin et al., 2017*; *Fessi et al., 2016*; *Iga et al., 2014*; *Kalapotharakos, Ziogas & Tokmakidis, 2011*; *Koundourakis et al., 2014*; *Lago-Peñas et al., 2013*; *Los Arcos et al., 2015*; *Meckel et al., 2018*; *Metaxas et al., 2006*; *Ostojic, 2003*; *Ostojic et al., 2009*; *Owen et al., 2018*; *Papadakis, Patras & Georgouli, 2015*; *Reinke et al., 2009*; *Silva et al., 2011*; *Suda et al., 2012*) and lean body mass (LBM, $n = 226$) (*Bonuccelli et al., 2012*; *Bunc, Hráský & Skalská, 2015*; *Casajus, 2001*; *D'Ascenzi et al., 2013*; *Devlin et al., 2017*; *Metaxas et al., 2006*; *Ostojic, 2003*; *Owen et al., 2018*; *Reinke et al., 2009*; *Suda et al., 2012*) are presented in Tables 1 and 2 and Figs. 1–3.

The overall analysis of a reasonable number of investigations seems to suggest that players' BM (Fig. 1) is stable during the season; trivial effects from PPS to BCP ($\Delta = -0.79\%$, ES $= -0.07$), MCP ($\Delta = -0.85\%$, ES $= -0.04$) and ECP ($\Delta = -1.33\%$, ES $= -0.12$) are examined by average.

Generally, both the absolute and relative BF decreases during the season (Figs. 1 and 2). From the observed studies, we might conclude that alterations of small magnitude are examined from PPS to BCP ($\Delta = -9.6\%$, ES $= -0.54$), MCP ($\Delta = -8.2\%$, ES $= -0.57$) and ECP ($\Delta = -8.7\%$, ES $= -0.39$) in absolute BF. In this line of study, relative BF may decrease by a small magnitude in BCP ($\Delta = 8.9\%$, ES $= 0.45$), MCP ($\Delta = 9.9\%$, ES $= 0.43$) and ECP ($\Delta = 12\%$, ES $= 0.53$). Interestingly, at BCP 88% (16 in 18), at 94% (MCP 17 in 18) and at ECP 100% (12 in 12) of the ES reported are negative and so pointing on a decrease in
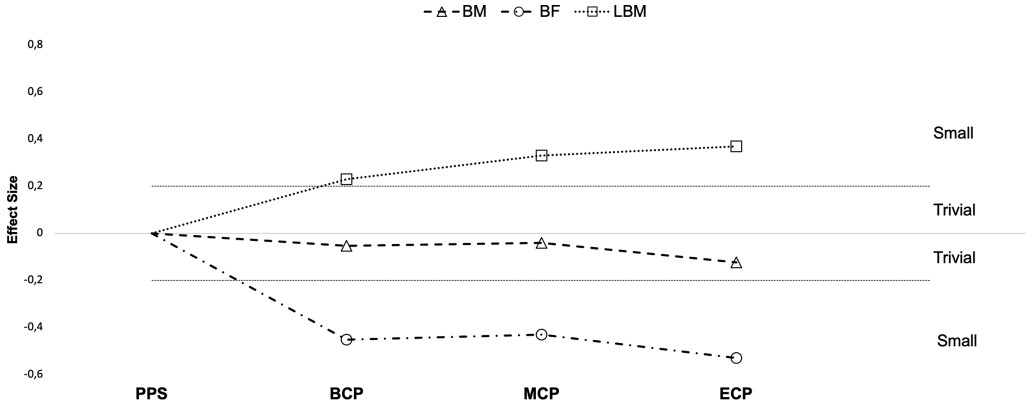

**Figure 1 Seasonal variations in body composition (average weighted effect sizes).** BM, body mass; BF, absolute and relative body fat; LBM, lean body mass; PPS, prior preseason phase; BCP, beginning competition phase; MCP, middle competition phase; ECP, end of competition phase.

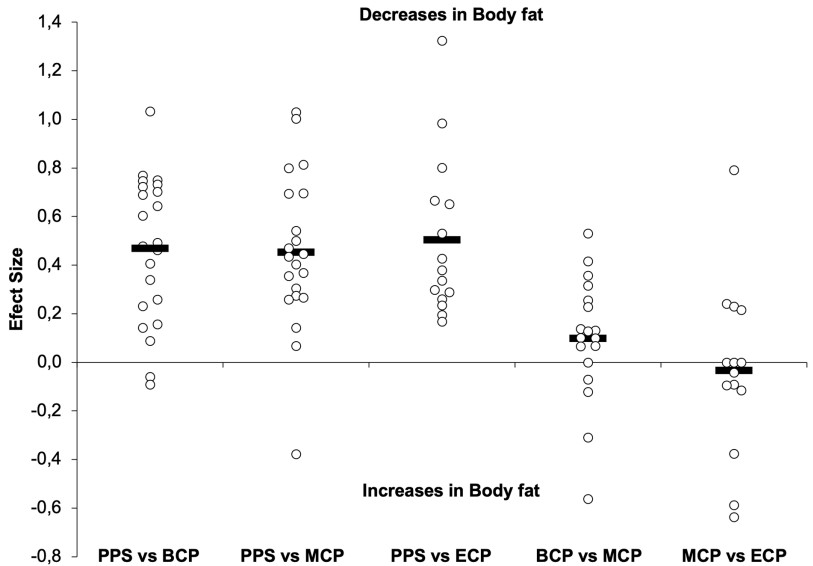

**Figure 2 Seasonal variations in absolute and relative body fat (weighted effect sizes).** PPS, prior preseason phase; BCP, beginning competition phase; MCP, middle competition phase; ECP, end of competition phase; dashed line represents average values.

absolute BF. Moreover, there are reports of decrements by moderate magnitude at BCP (*Clemente et al., 2021*; *D'Ascenzi et al., 2013*; *Devlin et al., 2017*; *Meckel et al., 2018*; *Ostojic, 2003*), MCP (*D'Ascenzi et al., 2013*; *Devlin et al., 2017*; *Kalapotharakos, Ziogas & Tokmakidis, 2011*; *Meckel et al., 2018*) and ECP (*D'Ascenzi et al., 2013*; *Koundourakis et al., 2014*; *Ostojic, 2003*; *Papadakis, Patras & Georgouli, 2015*). Although on average trivial changes in BF (absolute and relative) may occur during in-season (BCP *vs* MCP and MCP *vs* ECP, Δ = −2.4% and −1.8%, ES = −0.06 and −0.07, respectively), within the 16 studies that monitored in-season changes, both substantial decrements (*Casajus, 2001*; *D'Ascenzi et al., 2013*; *Fessi et al., 2016*; *Kalapotharakos, Ziogas & Tokmakidis, 2011*; *Koundourakis*

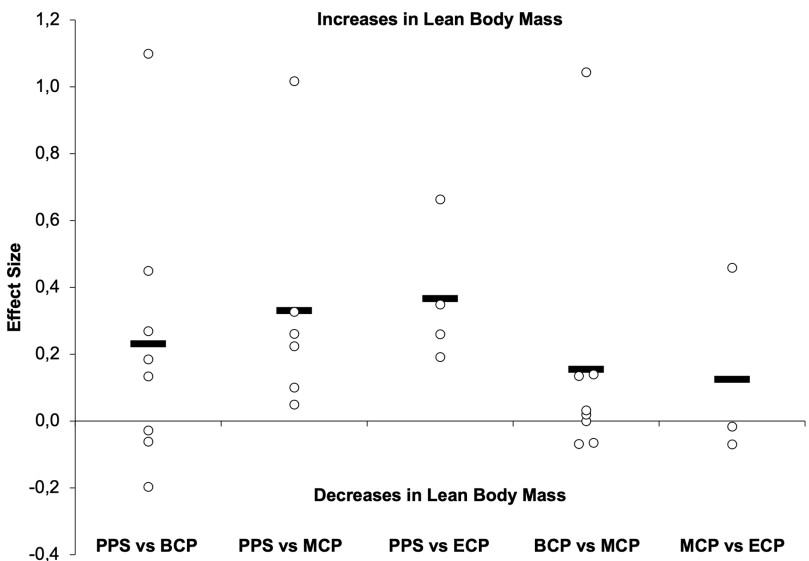

**Figure 3 Seasonal variations in lean body mass (weighted effect sizes).** PPS, prior preseason phase; BCP, beginning competition phase; MCP, middle competition phase; ECP, end of competition phase; dashed line represents average values.

*et al., 2014*; *Ostojic, 2003*; *Papadakis, Patras & Georgouli, 2015*; *Suda et al., 2012*) and increments (*Devlin et al., 2017*; *Papadakis, Patras & Georgouli, 2015*; *Suda et al., 2012*) are reported.

In this line of evidence towards a more positive body composition profile during season, the overall analysis of the studies suggest that players may substantially increase LBM during in-season. Our analyses reveal increases of small magnitude at BCP ($\Delta$ = 1.3%, ES = 0.23), MCP ($\Delta$ = 1.8%, ES = 0.33) and ECP ($\Delta$ = 3.1%, ES = 0.37) concerning PPS. Importantly, there are no reports of substantial decreases in LBM within the in-season assessments (BCP *vs* MCP and MCP *vs* ECP). On average increments of trivial magnitude are observed from BCP to MCP ($\Delta$ = 0.67%, ES = 0.15) and MCP to ECP ($\Delta$ = 1.2%, ES = 0.12). Curiously, variations in body composition seems to not be associated with the players' participation time (combined training and match exposure time) and did not differ across seasons (*Carling & Orhant, 2010*) and are independent of players position (*Milanese et al., 2015*). In summary, the general picture (Fig. 1) may suggest that professional players may maintain their BM after starting the training period through decreases in BF and increases in LBM. Although off-season detraining seems to reverse these anthropometric adaptations, with alterations of small magnitude in BM ($\Delta$ = 1.9%, ES = 0.2), BF ($\Delta$ = 1.6%, ES = 0.5) and decrements of moderate magnitude in LBM ($\Delta$ = 5%, ES = 0.9) (*Silva et al., 2016*) they may return to "optimal" initial values for competition after the preparation period. Factors related to training (*e.g.*, the type of strength training), competition fixtures (*e.g.*, extent of the pre-season and/or in-season period, mid-season breaks) and diet (*e.g.*, a Mediterranean diet) (*Ostojic, 2003*) may, among other factors, may explain part of the observed variability throughout the season (*e.g.*, BF). Nevertheless, the computed values for the different BM, BF and LBM were
derived from diverse assessment methods that have different measurements and precision errors associated (*Mills, De Ste Croix & Cooper, 2017*). Moreover, only a general picture has been provided and so, not capable to characterize the different body regions and associated seasonal variations.

## Force production

Longitudinal studies examining changes in the force production capacity of specific muscle groups in professional players mainly relied on isokinetic assessments, despite the discrepancy in the angular velocities analyzed (Table 2) (*Eniseler et al., 2012*; *Malliou et al., 2003*; *Silva et al., 2011*). Seasonal alterations in force production capabilities of specific muscles groups at angular velocities of $60°/s^{-1}$ (*Eniseler et al., 2012*; *Malliou et al., 2003*), $90°/s^{-1}$ (*Silva et al., 2011*), $180°/s^{-1}$ (*Malliou et al., 2003*), $300°/s^{-1}$ and $500°/s^{-1}$ (*Eniseler et al., 2012*) have been analyzed. Off-season induces alterations of small magnitude in knee extensors force production capacity at moderate ($180°/s^{-1}$) angular velocities (KE, $\Delta = 3.9\%$, ES = 0.37); no substantial alterations were observed at low angular velocities ($60°/s^{-1}$, $\Delta = -0.8\%$, ES = $-0.07$) (*Malliou et al., 2003*). During preseason, trivial effects are by average observed in KE at angular velocities of $60°/s$ (ranging from 227–272 and 222–229 N·m, respectively at PPS and BCP) (*Malliou et al., 2003*), $90°/s^{-1}$ (ranging from 239–242 and 241 N·m, respectively at PPS and BCP) (*Silva et al., 2011*) and $180°/s$ (ranging from 150–155 and 157–158 N·m, respectively at PPS and BCP) (*Malliou et al., 2003*). The same was observed for KF at $90°/s^{-1}$ (ranging from 129–131 and 129–132 N·m, respectively at PPS and BCP) (*Silva et al., 2011*). In this line of evidence, effects of trivial magnitude are by average observed from PPS to MCP for KE and KF, respectively when evaluated at $90°/s$ (KE, ranging from 241 N·m and KF ranging from 133–135 N·m at MCP) (*Silva et al., 2011*). Interestingly, when profiling adaptation in the force-velocity continuum perspective from PPS to ECP, small decrements at low (ranging from 272–273 and 251–253 N·m, respectively) (*Eniseler et al., 2012*), changes of trivial magnitude at moderate (ranging from 239–242 and 244 N·m, respectively) (*Silva et al., 2011*) and very large alterations at high angular velocities (ranging from 74–80 and 136–150 N·m, respectively) (*Eniseler et al., 2012*) for KE strength have been reported. Interestingly, a consistent substantial increment is KF force production from PPS to ECP seems to take place independently of the angular velocity evaluated. Specifically, from small magnitudes at low ($<60°/s^{-1}$, ranging from 148–150 and 159–178 N·m, respectively), moderate at moderate angular velocities ($90°/s^{-1}$, ranging from 129–131 and 134–138 N·m, respectively) and moderate and at high ($300°/s^{-1}$, ranging from 97–107 N·m, respectively) and very large at very high angular velocities ($500°/s^{-1}$, ranging from 148–150 and 159–178 N·m, respectively), respectively (*Eniseler et al., 2012*; *Silva et al., 2011*). This is particularly interesting, since is well documented that soccer-related injuries likely occur under rapid movement perturbations or actions requiring rapid force development and are more prevalent in hamstring muscles group (*Hagglund, Walden & Ekstrand, 2005*; *Walden et al., 2015*).

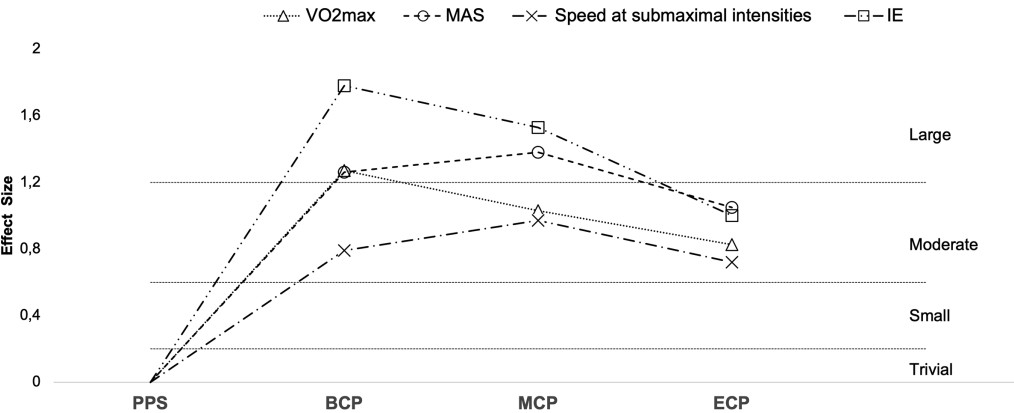

**Figure 4 Seasonal variations in neuromuscular performance (average weighted effect sizes).** CMJ$_{Based}$, single actions including countermovement (countermovement jump with and without arm swing); Non-CMJ, single actions not including a countermovement (squat jump with and without arm swing); ACC$_{Phase}$, acceleration phase (5-10-15 and 20 m distances); MV$_{Phase}$, maximal velocity phase (30 and 50 m distances); PPS, prior preseason phase; BCP, beginning competition phase; MCP, middle competition phase; ECP, end of competition phase.

## Jump ability

Seasonal changes in jump ability (15 studies, 390 players, Tables 1 and 2, Figs. 4–6) have frequently investigated the performance on single non-countermovement jump (Non-CMJ, SJ and SJWAS, Fig. 7) (*Aziz, Tan & Teh, 2005*; *Casajus, 2001*; *Koundourakis et al., 2014*; *Lago-Peñas et al., 2013*; *Malliou et al., 2003*) and single (CMJ$_{Based}$, CMJ and CMJWAS; Fig. 8) (*Casajus, 2001*; *Clark et al., 2008*; *Eliakim et al., 2018*; *Fessi et al., 2016*; *Koundourakis et al., 2014*; *Lago-Peñas et al., 2013*; *Los Arcos et al., 2015*; *Malliou et al., 2003*; *Meckel et al., 2018*; *Ostojic et al., 2009*; *Papadakis, Patras & Georgouli, 2015*; *Requena et al., 2017*; *Silva et al., 2011*), and repeated countermovement jumps (*Casajus, 2001*; *Clark et al., 2008*).

### Non-countermovement jump

The Non-CMJ$_s$ (Fig. 5) improves with a small magnitude during preseason training ($\Delta$ = 3.1%, ES = 0.27) but greater magnitudes can be observed by average from PPS to MCP ($\Delta$ = 7.8%, ES = 0.83) and ECP ($\Delta$ = 10%, ES = 1.04), respectively. Interestingly, at BCP 66% (one trivial (*Malliou et al., 2003*), small (*Lago-Peñas et al., 2013*) and moderate effect (*Aziz, Tan & Teh, 2005*)), at MCP (one small (*Lago-Peñas et al., 2013*) and large (*Aziz, Tan & Teh, 2005*) and three moderate effects (*Koundourakis et al., 2014*)) and at ECP (two moderate (*Koundourakis et al., 2014*) and two large (*Aziz, Tan & Teh, 2005*; *Koundourakis et al., 2014*)) 100% of the ESs calculated are substantial and so suggestive of an increase in non-CMJ ability. Although more scarcely investigated, trivial effects are by average computed between in-season assessments (BCP *vs* MCP and MCP *vs* ECP).

### Countermovement jump

Generally, CMJ$_{Based}$ (Fig. 6) improves by average with a small magnitude from PPS to BCP ($\Delta$ = 1.8%, ES = 0.26), MCP ($\Delta$ = 4.0%, ES = 0.47) and ECP ($\Delta$ = 3.3%, ES = 0.43). Interestingly, at BCP, 58% (seven in 12, six of small (*Los Arcos et al., 2015*; *Meckel et al.,*
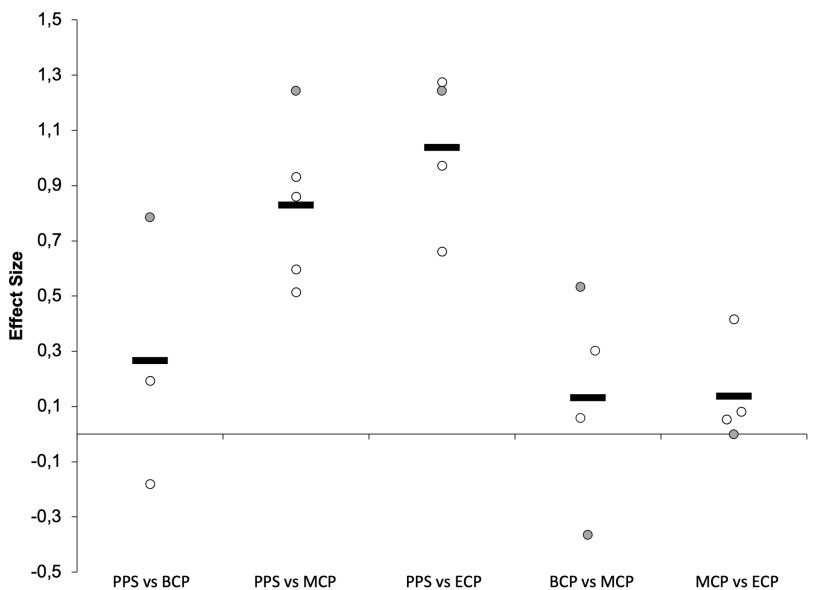

**Figure 5 Seasonal variations in non-countermovement jumps within the different studies (weighted effect sizes).** PPS, prior preseason phase; BCP, beginning competition phase; MCP, middle competition phase; ECP, end of competition phase; gray filled circles (squat jump with arm swing); white filled circles, squat jump without arm swing; dashed line represents average values. 

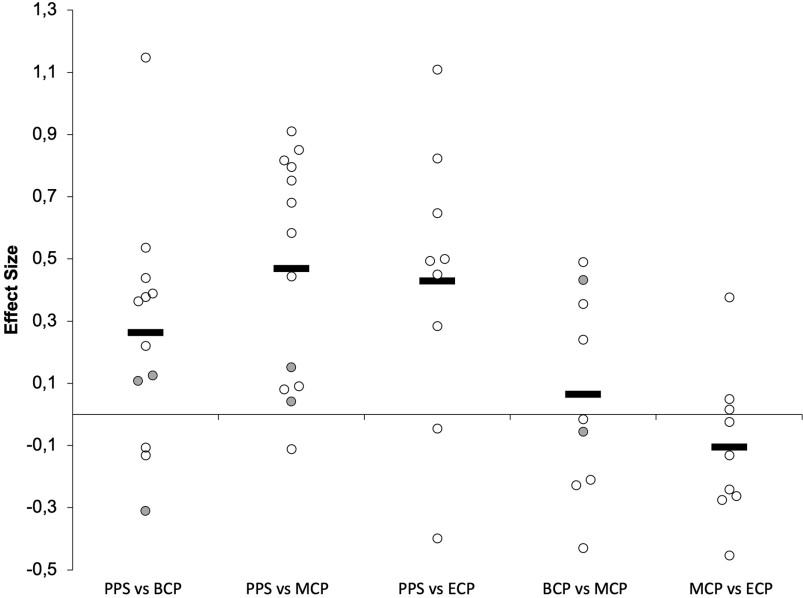

**Figure 6 Seasonal variations in countermovement-based jumps (weighted effect sizes).** PPS, prior preseason phase; BCP, beginning competition phase; MCP, middle competition phase; ECP, end of competition phase; gray filled circles (countermovement-jump with arm swing); white filled circles, countermovement-jump without arm swing; dashed line represents average values. 

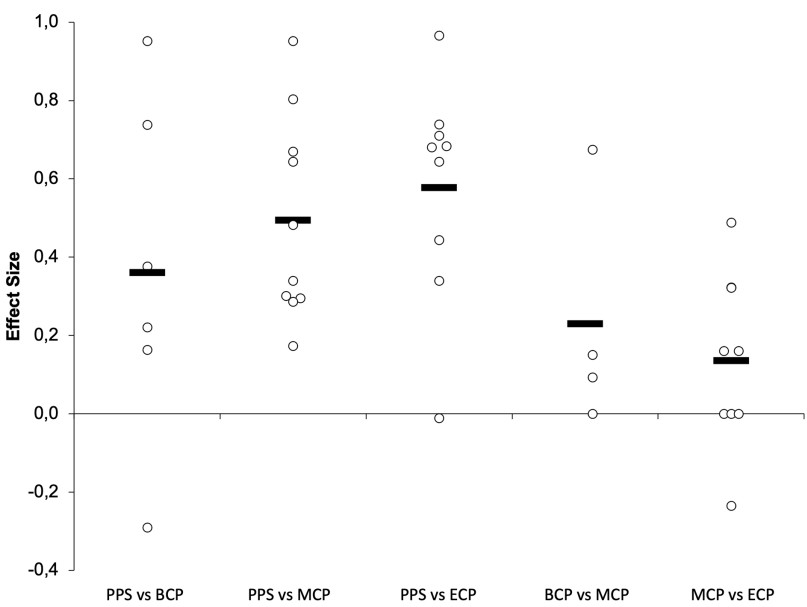

**Figure 7 Seasonal variations in the acceleration phase of the sprint (weighted effect sizes).** PPS, prior preseason phase; BCP, beginning competition phase; MCP, middle competition phase; ECP, end of competition phase; dashed line represents average values.

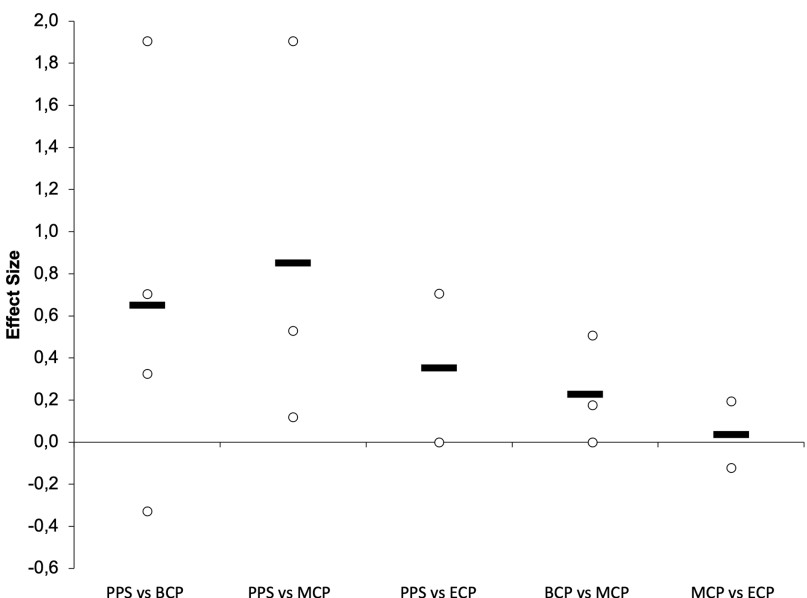

**Figure 8 Seasonal variations in the maximal velocity phase of the sprint (weighted effect sizes).** PPS, prior preseason phase; BCP, beginning competition phase; MCP, middle competition phase; ECP, end of competition phase; dashed line represents average values.

*2018*; *Papadakis, Patras & Georgouli, 2015*; *Silva et al., 2011*) and one of moderate magnitude (*Fessi et al., 2016*)), at MCP 61% (height in 13, two of small (*Koundourakis et al., 2014*; *Meckel et al., 2018*) and six of moderate magnitude (*Clark et al., 2008*; *Fessi et al., 2016*; *Koundourakis et al., 2014*; *Papadakis, Patras & Georgouli, 2015*)) and at ECP

78% (seven in nine, four of small (*Clark et al., 2008*; *Koundourakis et al., 2014*; *Papadakis, Patras & Georgouli, 2015*) and three of moderate magnitude (*Koundourakis et al., 2014*; *Papadakis, Patras & Georgouli, 2015*)) of the ES computed are indicative of an substantial increase in $CMJ_{Based}$ performance. Although more scarcely investigated, trivial effects are by average computed between in-season assessments (BCP *vs* MCP and MCP *vs* ECP).

Moreover, it seems that the maximal mechanical power and ability to sustain fatigue during the repeated performance of $CMJ_{Based}$ (average height during 20-s or mean power during a 15-s CMJ test) may improve with moderate and small magnitudes from PPS to MCP ($\Delta = 6.7\%$, ES = 0.74) and ECP ($\Delta = 2.2\%$, ES = 0.24), respectively (*Casajus, 2001*; *Clark et al., 2008*).

The improvements in jump ability after players return to normal training routines are somewhat expected as detraining or training cessation during off-season results in small to moderate decrements in jump ability ($\Delta = 4\%$ to 5.3%, ES = 0.4 to 0.8) (*Silva et al., 2016*). Importantly, we would like to call the reader attention for the variability in responses that can be observed during in-season. With this intention we select the performance of CMJ to expound on this problematic. Although CMJ return to "competition" values after players restart normal training routines, there are quite inconsistent responses during in season. In fact, the overall trivial effects from BCP (ranging from 37.5–55.8-cm) to MCP (ranging from 39.4–52.7-cm) are the result of three decrements (*Casajus, 2001*; *Fessi et al., 2016*; *Silva et al., 2011*) and increments of small magnitude (*Meckel et al., 2018*; *Papadakis, Patras & Georgouli, 2015*) and one trivial effect (*Lago-Peñas et al., 2013*). From MCP to ECP (ranging from 40.9–51.5-cm), the trivial changes are a product of the four decrements (*Clark et al., 2008*; *Papadakis, Patras & Georgouli, 2015*), one improvement of small magnitude (*Koundourakis et al., 2014*) and the four trivial effects examined (*Koundourakis et al., 2014*; *Requena et al., 2017*; *Silva et al., 2011*). All these investigations inform on the substantial variability in team responses that can be observed, and consequently, advise the practitioner on the need to provide players a consistent neuromuscular stimulus throughout the season. Although not universally confirmed, jump abilities may improve during the pre-season and can be further improved in-season when some mode of strength/power intervention is applied to the normal training routines of highly trained soccer players (*Allen et al., 2021*; *Silva, 2019*; *Silva, Nassis & Rebelo, 2015*). Given the large inter-individual variability of responses to training programs and match stimulus, efforts are being developed to optimize training programs at individual level (*Haugen, 2018*; *Jimenez-Reyes et al., 2022*; *Loturco et al., 2015a*; *Morin & Samozino, 2016*).

## Sprint ability
### Linear speed

Seasonal changes in the sprint (Tables 1 and 2, Fig. 4, $n = 230$ players) acceleration phase ($ACC_{Phase}$, 5 to 20-m distances, Fig. 7) (*Aziz, Tan & Teh, 2005*; *Fessi et al., 2016*; *Koundourakis et al., 2014*; *Los Arcos et al., 2015*; *Requena et al., 2017*; *Silva et al., 2011*) and maximal velocity phase ($MV_{Phase}$; 30 to 50-m, Fig. 8) (*Fessi et al., 2016*; *Ostojic, 2003*; *Silva et al., 2011*; *Zoppi et al., 2006*) has been analyzed. Traditionally, improvements in

sprint ability after players return to normal training routines are fairly expected as offseason results in moderate decrements (ES = 0.8 to 1.0) in $ACC_{Phase}$ (~2.5%) and $MV_{Phase}$ (~7%) (*Silva et al., 2016*). Specifically, when $ACC_{Phase}$ and $MV_{Phase}$ are distinctly examined, the later, although more sparsely investigated, tends generally to result in greater improvements and somewhat more substantial effects, as observed between:

- PPS (ranging from 0.97–1.04, 1.78–1.8, 1.83–2.29, 3.04–3.07, 4.16–4.9-s for 5, 10 15, 20 and 30-m sprint time) to BCP (Δ = 1.2% *vs* 3%; ES = 0.36 *vs* 0.65, respectively for $ACC_{Phase}$ and $MV_{Phase}$ (ranging from 0.95–1.06-s, 1.70-s, 2.27-s, 3.01-s, 4.22 to 4.7-s for 5, 10-, 15-, 20- and 30-m sprint time)).
- PPS to MCP (Δ = 1.9% *vs* 3.1%; ES = 0.49 *vs* 0.85, respectively for $ACC_{Phase}$ and $MV_{Phase}$ (ranging from 1.02, 1.70–1.76, 3.01–3.05, 4.14–4.7-s for 5, 10, 20 and 30-m sprint time)).
- PPS to ECP (Δ = 2.3% *vs* 3.3%; ES = 0.58 *vs* 0.35, respectively for $ACC_{Phase}$ and $MV_{Phase}$ (ranging from 1.0–1.03, 1.73–1.76, 2.95–3.04 and 4.16-s for 5, 10, 20 and 30-m sprint time)).

Curiously, within the season the variations and magnitudes are almost identical; BCP to MCP (Δ = 1% ES = 0.23) and MCP to ECP (Δ = 0.7% and ES = 0.04 *vs* 0.14, respectively for $ACC_{Phase}$ and $MV_{Phase}$). However, the phase analysis results in the inclusion of a reduced number of studies within each stage (more robustly in the $MV_{Phase}$) with obvious consequences in the interpretation of the results. Importantly, we would like to call the reader attention for the variability in the observed responses during in season. Moreover, recently examination of the force-velocity profiling during sprint reported that sprint mechanical properties are subjected to change during the season (*Haugen, 2018*; *Jimenez-Reyes et al., 2022*). Particularly, the theoretical maximal horizontal force production seems to be more compromised than maximal velocity towards the end of the season. Accordingly, the authors suggest that specific training stimuli should be consistently applied to increase maximal speed and acceleration (*Haugen, 2018*; *Jimenez-Reyes et al., 2022*).

### Change of direction speed

COD seems to be negatively affected during the offseason (Δ = 1.6%, ES = 0.6) (*Silva et al., 2016*). During preseason training players may restore their COD when evaluated by the time to perform a 4 × 10 m task (Δ = 2.5%, ES = 1.0) (*Meckel et al., 2018*). However, this was not observed when evaluated by the T-test (Δ = 0.5%, ES = 0.12) (*Silva et al., 2011*). Contradictory findings were also reported from PPS to MCP with trivial (Δ = 0.0%, ES = 0.0) (*Meckel et al., 2018*) and moderate (Δ = 3.5%, ES = 0.95) (*Silva et al., 2011*) improvements being reported simultaneously. However, they may stay consistent until ECP (Δ = 2.4%, ES= 0.67) when compared to PPS. Whitin the season both moderate performance decrements (Δ = 2.5%, ES = 0.78) (*Meckel et al., 2018*) and increments (Δ = 3.9%, ES = 1.0) (*Silva et al., 2011*) were recorded from BCP to MCP and a performance decrement of small magnitude from MCP to ECP was reported (Δ = 1.1%, ES = 0.32) (*Silva et al., 2011*).

## Insights from training

### *Preseason*

Pooled results from different experimental studies with professionals' players of different standards suggests that by average soccer players may experience a large (ES = ~1.25) increase in maximum dynamic strength performance during multi-joint exercises (~25% of $1RM_{Squat}$) throughout preseason training (*Silva, 2019*; *Silva, Nassis & Rebelo, 2015*). In fact, studies examining the effects of pre-season high-intensity strength training in force production, revealed that professional players improved maximum dynamic strength performance (1RM) in half-squat exercise (ranging from 11–26%) (*Bogdanis et al., 2009*; *Ronnestad et al., 2008*). The same evidence was observed following pre-season concurrent high-intensity aerobic and high-intensity strength training (~52%) (*Helgerud et al., 2011*). Moreover, improvements in relative force production (6–16%; LLV; 1RM/LLV) (*Bogdanis et al., 2009*) after high-intensity strength training and after concurrent high-intensity aerobic and high-intensity strength pre-season training are also reported (47%) (*Helgerud et al., 2011*).

Studies examining pre-season strength training programs reveal substantial improvements in, jump ability (5–10%) (*Bogdanis et al., 2009*; *Loturco et al., 2012*), acceleration (*Bogdanis et al., 2009*; *Loturco et al., 2012*), maximal speed phases (ranging from 1% to 2%) (*Bogdanis et al., 2009*; *Ronnestad, Nymark & Raastad, 2011*) and COD performance of profession al players (*Bogdanis et al., 2009*). More specifically, a ~23% and ~18% increase in IRM during Squat exercise may on average result in a ~7% and ~1.8% improvement in jump (CMJ and SJ) and sprint time (10 and 40 m) (*Silva, Nassis & Rebelo, 2015*). Nevertheless, improvements in jump ability and in maximal speed during preparation phase may be possibly associated with the type of strength training performed by players (weight training plus plyometric training vs weight training only) (*Ronnestad et al., 2008*). On the other hand, *Helgerud et al. (2011)* reported substantial improvements in CMJ (5%), and acceleration phase (1.6–3.3%) performance after pre-season concurrent high-intensity aerobic and strength training. Although already developed in the last millennium (*Tesch, Fernandez-Gonzalo & Lundberg, 2017*) and with "proof-of-concept" in soccer almost 20 years ago (*Askling, Karlsson & Thorstensson, 2003*), the systematic study of the training induced effects of isoinertial eccentric overload has been more recently implemented in soccer (*de Hoyo et al., 2015a*; *de Hoyo et al., 2015b*; *de Hoyo et al., 2016*; *Suarez-Arrones et al., 2018*; *Tous-Fajardo et al., 2016*). These previous studies reveal that this exercise model as shown to enhance common soccer tasks to at least a similar magnitude to those typical reported during the implementation of more traditional approaches during pre-season and in-season phases (*Allen et al., 2021*; *Silva, Nassis & Rebelo, 2015*).

### *In-season*

Regarding in-season alterations in strength parameters, *Ronnestad, Nymark & Raastad (2011)* observed that one high-intensity strength training session per week during the first 12-weeks of the in-season period was enough to maintain pre-season (2 week sessions throughout 10-weeks) gains in the strength performance of professional players. However,

a lower weekly in-season volume (one session every second week) only avoided the loss of training adaptations in jump performance; *i.e.*, strength and sprint performances decreased (*Ronnestad, Nymark & Raastad, 2011*). There are also reports of substantial improvements in 10-m (1.1%) fastest times during an RSA test of professional players after a periodized 4-week in-season specific high-intensity aerobic training intervention (*Owen et al., 2012*). These findings lead to the interesting hypothesis that strength-based actions present in SSG performance, *e.g.*, accelerations and decelerations, may stress the neuromuscular system to a point that allows in-season performance improvements in acceleration capacity (*Thomas, French & Hayes, 2009*). In fact, a high mechanical and metabolic load (acceleration/deceleration) seems to be imposed during soccer-specific scenarios (*Hodgson, Akenhead & Thomas, 2014*; *Osgnach et al., 2010*). As early mentioned, the different seasonal results during the performance of muscle-power-based efforts may be explained, at least in part, by the different neuromuscular stresses that are placed in players during the distinct periodization's applied by teams. Indeed, an extended longitudinal report (*Koundourakis et al., 2014*) tracking three professional teams suggest that squads who periodized training programs involving higher neuromuscular training loads during the season might show subsequent performance improvements throughout the seasonal continuum in both sprinting and jumping actions; differences in strength/power training stress between the analyzed teams were mainly due to the higher employed volume of both soccer-specific strength and sprint sessions performed by the different teams and not by the general resistance training contents. In this regard, soccer player programs should target all the force-velocity potential/spectrum of the neuromuscular system for a great transfer of this strength to sport activities; increasing player's ability to use strength and power effectively and consistently (*Silva, Nassis & Rebelo, 2015*). In fact, each player needs an individually optimized approach; one may need to prioritize the development of maximal force capabilities while others maximal velocity capabilities (*Morin & Samozino, 2016*). Moreover, adaptations at the neuromuscular level seem to not only be affected by training but also by the time of match exposure of the players (*Morgans, Di Michele & Drust, 2017*; *Silva et al., 2011*; *Sporis et al., 2011*). Despite the wide range of sprint distances evaluated, sprint ability may improve throughout the pre-season and further in-season and those improvements may be more marked during the acceleration phase (*Silva, Nassis & Rebelo, 2015*). The latest evidence is even more curious taking into consideration that analysis of games performed by young elite players (*Mendez-Villanueva et al., 2011*) reveals that athletes may rarely reached their maximal sprint speed during the game. Nevertheless, these was not observed in adult semi-professional players (*Massard, Eggers & Lovell, 2017*) and from our knowledge as not yet been investigated in high-level adult players.

In conclusion, the implementation of strength training routines as shown to result in increases of moderate magnitude in jump, linear speed (acceleration and maximal speed phases) and COD (*Silva, 2019*; *Silva, Nassis & Rebelo, 2015*). Moreover, the magnitude of adaptation and the training efficiency (% improvement by session) may be influenced by the chronic biomechanical and physiological context of the training program (*Loturco et al., 2015b*; *Silva, Nassis & Rebelo, 2015*). In fact, programs with greater biomechanical

specificity (*e.g.*, force being applied in all the velocity continuum and planes of motion) seems to result is greater improvements in the performance of the analyzed tasks (*Silva, Nassis & Rebelo, 2015*). Moreover, the physiological demands of the overall session organization (*e.g.*, degree of stress placed at the aerobic system) may affect the magnitude of adaptations (*Loturco et al., 2015a*; *Silva, 2019*). As example, research investigating the training induced effects of concurrent training programs observed that this training programs may produce increases of moderate magnitude in jump (~5.6%), sprint (3.2%) and COD (2.6%) (*Silva, 2019*). A systematic analysis suggest that greater magnitudes of adaptation and training efficiency scores can be detected when the physiological type of the session are more unidirectional (mechanical and metabolic sessions are performed alternatively) than multidirectional (strength and endurance in the same session) (*Silva, 2019*). In fact, the former organization seems to result in more substantial magnitudes of increases (moderate to large) in jump and sprint abilities that the later organization mode (small to moderate). When this is the case, adaptations may vary according to the session arrangement (endurance + strength and vice versa). Nevertheless, this systematic analysis included studies with professional and semi-professional players (*Silva, 2019*). According to *Silva (2019)* practitioners should adopt a holistic approach when defining the exercise timing of the strength-based component of the session. A couple of them are: (i) Is the player returning from injury or not? (ii) What is the training priority within this exact training period? (iii) Is the team in a congestive schedule period or not? (iv) What is the supposed metabolic/mechanical stress of the "overall" session? (iv) Does the player show enough technical competency to perform a complex strength exercise in fatigued state? (*Silva, 2019*). In summary, during the preparation phase players "recover" body composition and neuromuscular competitive capacity. Generally, improvements of small magnitude in non-CMJ and CMJ-based jumps and the acceleration ($ACC_{Phase}$) and maximal velocity phase ($MV_{Phase}$) of the sprint are observed from PPS to BCP. In the middle of competition phase, they are observed small (CMJ-based and $ACC_{Phase}$), and moderate (non-CMJ and $MV_{Phase}$) improvements compared to PPS. However, alterations towards end of competition phase seem to be force-velocity dependent; CMJ-based and $MV_{Phase}$ improve to a small extent with non-CMJ and sprint $ACC_{Phase}$ revealing moderate increments compared to PPS. Trivial alterations occur withing in-season in these parameters. However, these is the result of the variability observed between studies; more evident when monitoring the CMJ performance. Different resistance training methods or combination of methods may improve (pre-season) and assist in the maintenance or further improvement (in-season) of physiological determinants and neuromuscular performance during the season.

## ENDURANCE: WHY THE RELEVANCE?

Activity pattern analysis of the players during the matches showed that elite soccer players cover 8 to 13-km during a competitive match (*Bradley et al., 2009*; *Di Salvo et al., 2009*; *Rampinini et al., 2007b*) at a mean intensity close to the anaerobic threshold (AT) (*Stolen et al., 2005*). Moreover, energy expenditure during a match play averages 70–75% of the maximal oxygen consumption ($VO_{2max}$), which suggests that a high level of physical

performance in soccer may, in part, be determined by aerobic fitness (*Bangsbo, Mohr & Krustrup, 2006*; *Reilly & Ekblom, 2005*).

The determination of $VO_{2max}$ and AT are two of the most frequent parameters used when monitoring aerobic fitness in the laboratory settings. In addition, seasonal changes in the fitness of soccer players have also been examined by records of time to exhaustion (TE) and maximal aerobic speed (MAS) during maximal incremental tests performed in the laboratory or in field conditions. Although, the power of $VO_{2max}$ to discriminate higher and lower-level players have not been unanimously reported (*Marcos, Koulla & Anthos, 2018*; *Rampinini et al., 2009a*; *Slimani et al., 2019*; *Tonnessen et al., 2013*; *Wells et al., 2012*; *Ziogas et al., 2011*), higher values of $VO_{2max}$ have been positively associated with players in specific team position roles (midfielders) (*Tonnessen et al., 2013*). A better cardiovascular capacity, measures by means of $VO_{2max}$ and MAS seems related to a lower perception of exercise intensity during trainings and games (*Azcarate et al., 2020*). Moreover, players with poor aerobic fitness (*Malone et al., 2018*) or showing lower improvements during specific phases of the season (preseason) may have a greater risk of injury than players with better-developed aerobic fitness (*Eliakim et al., 2018*).

AT is defined as the highest exercise intensity, heart rate (HR) or $VO_2$, in which the production and clearance of lactate is equal (*Stolen et al., 2005*). Several methods exist to determine AT, including blood lactate and ventilatory measurements. Lactate threshold (LT) and ventilatory threshold (VT) have been advocated as more sensible physiological parameters to detect changes in the fitness of soccer players, rather than $VO_{2max}$ (*Clark et al., 2008*; *Edwards, Macfadyen & Clark, 2003*; *Helgerud et al., 2001*); velocity at LT can better discriminate endurance characteristics of soccer teams of different level (*Ziogas et al., 2011*). Moreover, LT might change without changes to $VO_{2max}$, and a higher LT means, theoretically, that a player can maintain a higher average intensity in an activity without the accumulation of lactate (*Helgerud et al., 2001*) and so, for the same external loads a lower internal homeostatic disturbance.

To increase the ecological validity of the measurements, maximal and sub-maximal soccer-specific field tests have been widely used to monitor the training status of professional soccer players. Recent evidence suggests that the intermittent endurance capacity of players is improving over time (*Elferink-Gemser et al., 2012*). Moreover, the level of competitiveness of the player is related to the performance in: (i) soccer-specific endurance tests, such as the 30-15 and the Yo-Yo tests (*Casado Yebras et al., 2014*; *Ingebrigtsen et al., 2012*; *Mohr, Krustrup & Bangsbo, 2003*; *Rampinini et al., 2009a*; *Wells et al., 2012*), (ii) repeated sprint ability tests with (RSSA) (*Rampinini et al., 2009b*; *Wells et al., 2012*) or without (RSA) (*Aziz et al., 2008*) changes of direction and (iii) to the intermittent exercise performance during games (*Mohr, Krustrup & Bangsbo, 2003*). Additionally, a positive relationship was observed between team success in the league and the Yo-Yo intermittent endurance test level 2 (YYIE2) (*Randers, Rostgaard & Krustrup, 2007*) and the Yo-Yo intermittent recovery test level 2 (YYIR2) (*Ingebrigtsen et al., 2012*). Several studies reported significant correlations between the performance on distinct intermittent endurance field tests and other physiological and performance measurements, such as $VO_{2max}$ (*Castagna et al., 2006*; *Jones et al., 2013*; *Krustrup et al., 2006*; *Rampinini*
*et al., 2009a*; *Rampinini et al., 2009b*; *Stanković et al., 2021*; *Wells et al., 2014*), $VO_2$ kinetics during high-speed running (HSR) (*e.g.*, velocity at 80%$\Delta VO_2$) (*Wells et al., 2014*), incremental treadmill test performance (ITT) (*Krustrup et al., 2006*), and TE during a maximal anaerobic running test (*Wells et al., 2014*). Moreover, improvements in the YYIR2 were associated with increases in power, TE and maximal speed during a maximal anaerobic running test (*Wells et al., 2014*). Importantly, YYIR1 (*Krustrup et al., 2003*), YYIE2 (*Bradley et al., 2010*) and in RS(S)A (*Altmann et al., 2018*; *Rampinini et al., 2007b*) performance have been shown to be associated with game-related physical activity (*e.g.*, total distance, HSR and sprint) (*Altmann et al., 2018*; *Bradley et al., 2010*; *Krustrup et al., 2003*; *Krustrup et al., 2005*; *Rampinini et al., 2007a*). Additionally, correlations between changes in intermittent endurance field tests and changes in match activity (*e.g.*, HSR) during the season, which were not evident for $VO_{2max}$ have been reported (*Bradley et al., 2011*). However, contradictory findings regarding measures of proficient match activity (HSR) have also been reported to correlate with laboratory fitness measures (ITT and $VO_{2max}$) (*Impellizzeri et al., 2006*; *Krustrup et al., 2003*; *Krustrup et al., 2005*). Nevertheless, it is important to note that most of the studies only detected moderate correlations and thus cannot be used to establish a direct cause-effect relationship (*Rampinini et al., 2007a*). Nevertheless, a greater discriminatory validity has been attributed to the field monitoring techniques and thus, at least in part, makes them more important (specific) in monitoring soccer players (*Buchheit, 2010*; *Ingebrigtsen et al., 2012*; *Svensson & Drust, 2005*; *Wells et al., 2012*). In addition, the examined reliability and sensitivity to training of field derived sub-maximal HR measures make these measures an important parameter for frequent, time-efficient, and non-exhaustive testing of intermittent exercise capacity of high-level soccer players (*Altmann et al., 2021*; *Buchheit, 2014*; *Buchheit, Simpson & Lacome, 2020*; *Ingebrigtsen et al., 2013a*; *Rago et al., 2020*). In fact, in addition to better physiological responses being observed in players of a higher standard (*Ingebrigtsen et al., 2012*), they also seem to be associated with acute (*Rago et al., 2020*) and chronic physical match performance (HSR) (*Bradley et al., 2010*). Notwithstanding the previous facts, it is important to highlight those maximal (*e.g.*, $VO_{2max}$ and MAS) and sub-maximal aerobic fitness laboratory parameters (*e.g.*, AT) cannot be neglected. In fact, others (*Altmann et al., 2018*; *Vincenzo, Franco & Carlo, 2013*) observed the ecological validity of these parameters *via* their association with match categories of an aerobic and anaerobic nature. As so, practitioners should always consider a cost/benefit approach (*e.g.*, cost, ease of use, manpower and how it will impact the training program) (*Buchheit & Simpson, 2017*).

## VARIATIONS IN PHYSIOLOGICAL DETERMINANTS AND ENDURANCE PERFORMANCE

### Maximal oxygen consumption

Although with obvious limitations, *e.g.*, just one study involves a longitudinal, inter-seasonal examination of soccer players (12 seasons, 1,545 players), it seems that among professional players, $VO_{2max}$ is not improving over time and perhaps has the

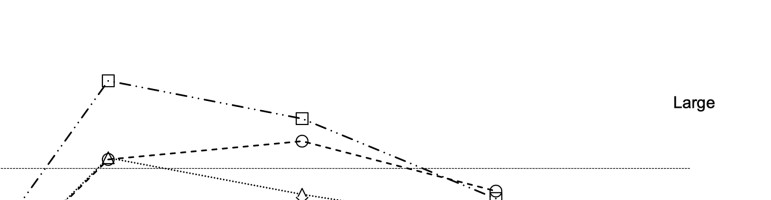

**Figure 9 Seasonal variations in physiological determinants and endurance performance (average weighted effect sizes).** $VO_{2Max}$, maximal oxygen consumption; Speed at sub-maximal intensities-speed recorded at blood lactate concentrations of 2 and 4 $mmol^{-1}$; MAS, maximal aerobic speed; IE, high-intensity intermittent exercise (30–15 and YO-YO tests); PPS, prior preseason phase; BCP, beginning competition phase; MCP, middle competition phase; ECP, end of competition phase.

tendency to decrease (players tested from 2006–2012 showed 3.2% lower values than those tested from 2000–2006) (*Tonnessen et al., 2013*).

Seasonal alterations in $VO_{2max}$ have been extensively analyzed (Table 2, Figs. 9 and 10, $n = 393$) (*Aziz, Tan & Teh, 2005*; *Bunc, Hráský & Skalská, 2015*; *Casajus, 2001*; *Castagna et al., 2013*; *Clark et al., 2008*; *Edwards, Macfadyen & Clark, 2003*; *Eliakim et al., 2018*; *Haritodinis et al., 2004*; *Kalapotharakos, Ziogas & Tokmakidis, 2011*; *Koundourakis et al., 2014*; *Lago-Peñas et al., 2013*; *Manzi et al., 2013*; *Meckel et al., 2018*; *Metaxas et al., 2009*; *Michalczyk et al., 2008*; *Mohr, Krustrup & Bangsbo, 2002*). Generally, during the pre-season, professional players appear to regain their oxygen capacity and maintain it throughout the season as off-season seems to induce a large impairment in this physiological parameter ($\Delta = 4.4\%$, ES = 1.4) (*Silva et al., 2016*). Studies with players from different backgrounds exposed a large magnitude of improvements in $VO_{2max}$ from the PPS (ranging from 52.2–62.7 $ml \cdot min^{-1} \cdot kg^{-1}$) to BCP ($\Delta = 7.3\%$, ES = 1.3, ranging from 54.8–66.5 $ml \cdot min^{-1} \cdot kg^{-1}$). Additionally, improvements of moderate magnitude were by average observed in the MCP ($\Delta = 6.4\%$, ES = 1.0, ranging from 55.5–66.8 $ml \cdot min^{-1} \cdot kg^{-1}$) and ECP ($\Delta = 4.2\%$, ES = 0.8 ranging from 52.7–64.1 $ml \cdot min^{-1} \cdot kg^{-1}$) compared with the PPS. Moreover, increases in $VO_{2max}$ (from PPS to BCP, MCP and ECP) seem to be independent of the position role (*Metaxas et al., 2006*). Interestingly, just one of the 14 ES did not confirm the substantial improvements at MCP and four on 13 at ECP compared to PPS assessments. Within the competitive phase they are observed by average trivial changes from BCP to MCP ($\Delta = 0.5\%$, ES = 0.1) and a small decrement from MCP to ECP ($\Delta = -2.3\%$, ES = -0.28).

## Anaerobic threshold

Studies examining changes in physiological parameters at sub-maximal intensities are presented in Table 2 (13 studies, $n = 249$, Fig. 9) (*Casajus, 2001*; *Castagna et al., 2011*; *Clark*

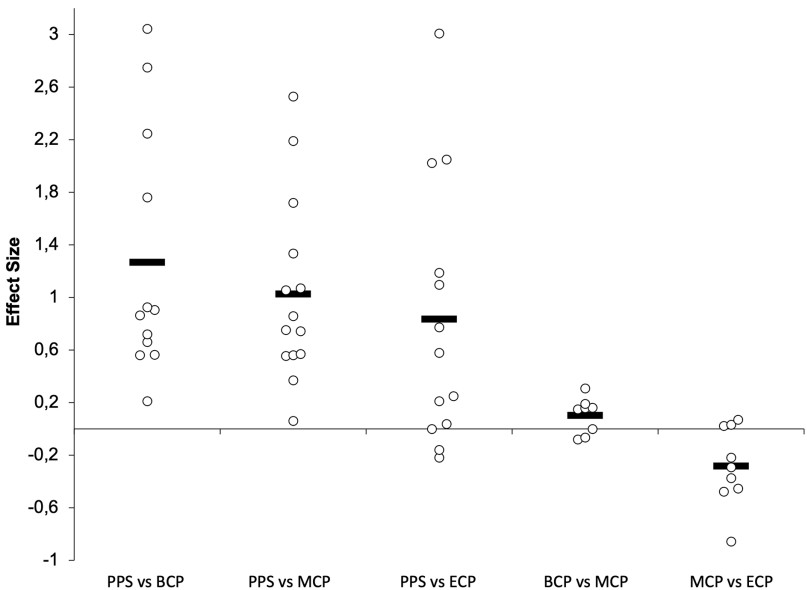

**Figure 10 Seasonal variations in VO$_{2Max}$ (weighted effect sizes).** PPS, prior preseason phase; BCP, beginning competition phase; MCP, middle competition phase; ECP, end of competition phase; dashed line represents average values.

*et al., 2008*; *Dunbar, 2002*; *Edwards, Macfadyen & Clark, 2003*; *Kalapotharakos, Ziogas & Tokmakidis, 2011*; *Los Arcos et al., 2015*; *Manzi et al., 2013*; *Meckel et al., 2018*; *Mohr, Krustrup & Bangsbo, 2002*; *Papadakis, Patras & Georgouli, 2015*; *Zoppi et al., 2006*).

From the large variety of parameters examined, some were shown to be sensitive in one but not in other studies that used players of similar standards. Nevertheless, enhancements in the ability to cope with sub-maximal internal and external loads regarding PPS performances were by average detected by different parameters, as follows:

i) the percentage of VO$_{2max}$ (76%VO$_{2max}$) and percentage of maximal heart rate (87% HR$_{max}$) at a lactate concentration of 4 mmol$^{-1}$ at BCP (ES = 0.62 and 0.71, 78% and 89%) and MCP (ES = 0.89 and 0.91, 78% and 89%), respectively) (*Kalapotharakos, Ziogas & Tokmakidis, 2011*);

ii) oxygen consumption at the LT (ES = 0.5 at ECP, ranging from 51.4–53.5 ml·min$^{-1}$·kg$^{-1}$) (*Edwards, Macfadyen & Clark, 2003*) and VT (ES = 0.85 and 0.41 at BCP (ranging from 50.2-52.7 ml·min$^{-1}$·kg$^{-1}$) and ECP (52.9 ml·min$^{-1}$·kg$^{-1}$), respectively) (*Casajus, 2001*; *Edwards, Macfadyen & Clark, 2003*; *Manzi et al., 2013*);

iii) heart rate measures at speeds of 14-km/h (ES = 2.7), 16-km/h (ES = 2.6), and 18-km/h (ES = 2.0) at MCP (*Mohr, Krustrup & Bangsbo, 2002*);

iv) the speed at a fixed lactate concentration (Fig. 9) of: (a) 2 mmol$^{-1}$ (ES = 0.67, 0.66 and 0.68, at BCP (ranging from 11.4–14.5-km/h), MCP (ranging from 10.5–14.8-km/h), and ECP (ranging from 10.8–13.9-km/h) regarding PPS (ranging from 9.5–14.3-km/h) respectively) (*Castagna et al., 2011*; *Castagna et al., 2013*; *Dunbar, 2002*; *Kalapotharakos, Ziogas & Tokmakidis, 2011*; *Manzi et al., 2013*; *Papadakis, Patras & Georgouli, 2015*); (b) 3 mmol$^{-1}$ (ES = 0.52, 0.20 and −0,27 at BCP (ranging from

12.7–15.4-km/h), MCP (15.7 km/h), and ECP (15.0-km/h) regarding PPS (ranging from 12.2–15.4-km/h), respectively) (*Dunbar, 2002*; *Los Arcos et al., 2015*); (c) 4 mmol$^{-1}$ (ES = 1.0, 1.41 and 1.27 at BCP (ranging from 13.6–14.9-km/h), MCP (ranging from 13.6–14.4-km/h) and ECP (ranging from 13.5–14.3-km/h), regarding PPS (ranging from 12.3–13.9-km/h), respectively) (*Castagna et al., 2011*; *Castagna et al., 2013*; *Kalapotharakos, Ziogas & Tokmakidis, 2011*; *Manzi et al., 2013*; *Papadakis, Patras & Georgouli, 2015*);

v) The speed at the LT (ES = 1.9 at BCP, ranging from 10.5–13.8-km/h, respectively) (*Zoppi et al., 2006*) and VT (ES = 0.57 and 1.1, at BCP (ranging from 11.6–12.2-km/h) and MCP (12.8-km/h), respectively) (*Meckel et al., 2018*).

Interesting, although a wide variety of submaximal parameters have been measured, substantial improvements between BCP and MCP are consistently reported within the analyzed studies in some form of physiological parameter (*Casajus, 2001*; *Dunbar, 2002*; *Kalapotharakos, Ziogas & Tokmakidis, 2011*; *Meckel et al., 2018*; *Papadakis, Patras & Georgouli, 2015*). Nevertheless, alterations of trivial magnitude have also been examined (*Kalapotharakos, Ziogas & Tokmakidis, 2011*; *Papadakis, Patras & Georgouli, 2015*). From the MCP to ECP distinct alterations have been observed with both reports of trivial (*Papadakis, Patras & Georgouli, 2015*), small improvements (*Papadakis, Patras & Georgouli, 2015*) and impairments (*Dunbar, 2002*).

Interestingly, although trivial alterations in $VO_{2max}$ are by average observed from BCP to MCP, an improvement of small magnitude (ES = 0.29) is observed between these time points, which suggest that further improvement in sub-maximal exercise performance (*e.g.*, LT), but not in $VO_{2max}$, are likely related to a faster restoration or improvement of central factors (*i.e.*, $VO_{2max}$) than peripheral factors (*i.e.*, muscle oxidative enzymes) (*Impellizzeri et al., 2006*). Furthermore, although adaptations in RE being dependent on multi-dimensional factors (*e.g.*, mechanical, and neuromuscular skills) they may had occurred further in season, and so determinant for improving running performance (*Foster & Lucia, 2007*); RE can better discriminate soccer players of different standards with similar $VO_{2max}$ values (*Ziogas et al., 2011*).

In summary, these physiological determinants of endurance performance, improve during the first part of the season (4–8 weeks) and generally remain stable throughout the season. Generally, improvements in $VO_{2max}$ occurred after a relatively short period of time (*e.g.*, pre-season training), while no significant further in-season increases are observed. Moreover, no increase was examined in the $VO_{2max}$ when players already possessed values of approximately 61–62 ml/kg/min. In fact, the increases in $VO_{2max}$ found in different standard of players during the in-season period (*Caldwell & Peters, 2009*; *Jensen et al., 2009*; *Magal et al., 2009*) occurred under this threshold and in players of a lower standard. Additionally, when professional players began the competitive season with values above this threshold (61–62 ml/kg/min), no improvements in the $VO_{2max}$ throughout the season were by average reported (*Clark et al., 2008*; *Edwards, Macfadyen & Clark, 2003*). This may be related with soccer training-specific constrains and/or demands, such as, the limited time for fitness training due to the high density of in-season match commitments.

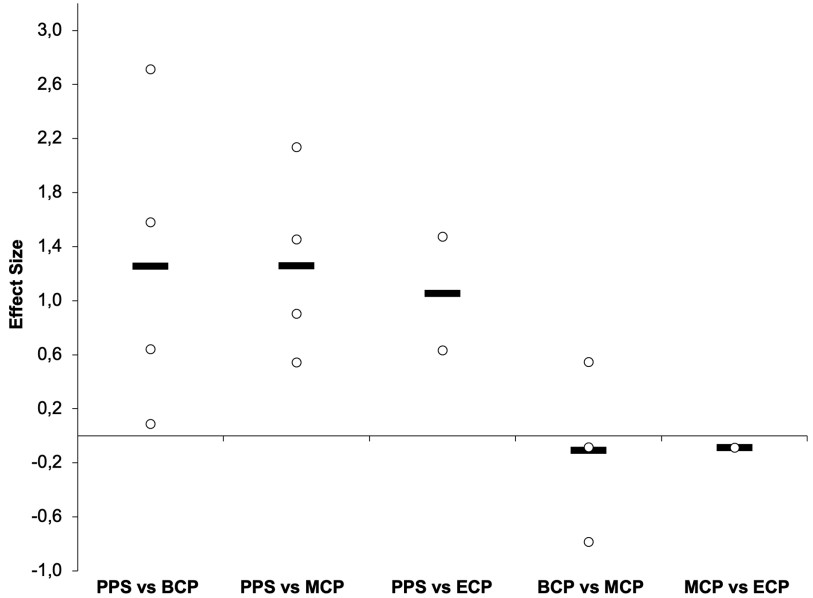

**Figure 11 Seasonal variations in maximal aerobic speed (weighted effect sizes).** PPS, prior preseason phase; BCP, beginning competition phase; MCP, middle competition phase; ECP, end of competition phase; Dashed line represents average values.

Our analysis seems to corroborate the observations of others (*Tonnessen et al., 2013*), indicating that $VO_{2max}$ values of approximately 62–64 ml/min/kg may fulfill the general demands for aerobic capacity in male professional soccer players; nevertheless, characteristics related to the specific demands of different positional roles should be considered, as reported values reflect team averages and large inter-individual variations can be observed.

## Maximal aerobic speed

MAS (Figs. 9 and 11) ($n = 143$) (*Boullosa et al., 2013*; *Bunc, Hráský & Skalská, 2015*; *Fessi et al., 2016*; *Kalapotharakos, Ziogas & Tokmakidis, 2011*; *Lago-Peñas et al., 2013*; *Requena et al., 2017*) reflects the maximum aerobic capacity and combines $VO_{2max}$ and RE into a single factor (*Billat & Koralsztein, 1996*). As such, MAS is a good indicator of aerobic performance (*Billat & Koralsztein, 1996*), and the determination of MAS gives a practical assessment of the aerobic demands during running performance (*Kalapotharakos, Ziogas & Tokmakidis, 2011*). Off-season break may induce a decrement of moderate magnitude in MAS ($\Delta = 4.6\%$, ES = 0.61) (*Requena et al., 2017*). Preseason training restores MAS ($\Delta = 5\%$, ES = 1.3, ranging from 18.1–19.7-km/h), with substantial improvements still evident at MCP ($\Delta = 4.3\%$, ES = 1.3, ranging from 17.4–19.6-km/h) and at ECP ($\Delta = 4.9\%$, ES = 1.05, ranging from 17.3–18.4-km/h) regarding the PPS values (ranging from 16.5–19.2-km/h). Although, by average no substantial improvements take place from BCP to MCP ($\Delta = -0.7\%$, ES = $-0.11$) and from MCP to ECP ($\Delta = 0.6\%$, ES = $-0.09$), there are contradictory observations between BCP and MCP, with both trivial ($\Delta = -0.4\%$, ES = $-0.09$) (*Lago-Peñas et al., 2013*), moderate impairments ($\Delta = -3.3\%$, ES = $-0.82$) (*Fessi et al., 2016*) and improvements of small magnitude ($\Delta = 1.7\%$, ES = 0.59)

(*Kalapotharakos, Ziogas & Tokmakidis, 2011*) reported. Interestingly, *Boullosa et al. (2013)* did not observed changes in the MAS (18.1 to 18.2-km/h) in professional players after pre-season. The different findings are, at least in part, associated with the dissimilar baseline MAS that were reported and the applied protocols (*Dupont, Akakpo & Berthoin, 2004*; *Fessi et al., 2016*; *Kalapotharakos, Ziogas & Tokmakidis, 2011*; *Wong et al., 2010*). We would like to highlight that in this narrative review we discussed the velocity at $VO_{2max}$ ($vVO_{2max}$) (*Kalapotharakos, Ziogas & Tokmakidis, 2011*), and final velocity reached (Vam-eval and Gacon test) as one parameter (*Boullosa et al., 2013*; *Bunc, Hráský & Skalská, 2015*; *Fessi et al., 2016*; *Lago-Peňas et al., 2013*; *Requena et al., 2017*). Although they are highly correlated, with the two terms being often used interchangeably, they refer to different physiological entities (*Buchheit, 2010*) with MAS maybe 10–15% greater than the $vVO_{2max}$ (*Berthon & Fellmann, 2002*).

In summary, despite the scarcity of research monitoring these performance parameters, MAS increase after pre-season training and remain stable throughout the season. The magnitude of alterations (MAS) may be associated with the baseline training status of players at the time of intervention (*Boullosa et al., 2013*).

## High-intensity intermittent exercise

A summary of studies examining changes in high intensity intermittent exercise (IE) tests is presented in Tables 1 and 2 and Figs. 9 and 12 (*Boullosa et al., 2013*; *Bradley et al., 2010*; *Campos-Vazquez et al., 2016*; *Castagna et al., 2013*; *Iaia et al., 2009b*; *Krustrup et al., 2003*; *Krustrup et al., 2006*; *Manzi et al., 2013*; *Silva et al., 2011*). Off-season seems to result in decrements of moderate and very large magnitude in IE performance ($\Delta$ = 27.8% and 10%, ES = 1.0 and 2.2 for YYIE2 and YYIR2, respectively). However, preseason phase by average induces large improvements IE ($\Delta_{overall}$ = 32.4%, ES = 1.8). Specifically, improvements of 56%, 60%, 18% and 5%, and effect sizes of 4.1, 2.4, 1.1 and 1.25 for YYIR2 (ranging from 742–780-m and 1,033–1,160-m), YYIE2 (ranging from 1,120–2,171-m and 2,250–2,411-m), YYIR1 (ranging from 1,760–2,475-m and 2,211–2,600-m) and 30–15 (20.1 to 21.1 km/h), respectively. These performance improvements are extended to MCP ($\Delta_{overall}$ = 18.9%, ES = 1.5). Precisely, increases of 43.9%, and 17.9%, with magnitudes of 2.4 and 0.7 for YYIR2 (ranging from 742–780-m) and YYIE2 (ranging from 742–780-m). Interestingly, the magnitude of alterations is lower from PPS to ECP ($\Delta_{overall}$ = 22.5%, ES = 1.0). Specifically, increments of 11.9%, 29.7% and 19.5% with magnitudes of 0.51, 0.96 and 1.56, for YYIR2 (873-m), YYIE2 (ranging from 1,640–2,381-m), YYIR1 (2,103-m), are examined. Within the season, the ability to perform IE is by average impaired to a small extent from BCP to MCP ($\Delta_{overall}$ = −2.4%, ES = −0.23 ($\Delta$ = −7.2% and 6.1%, ES = −0.47 and 0.24 for YYIR2 and YYIE2, respectively)) and from MCP to ECP ($\Delta_{overall}$ = −7%, ES = −0.3). We would like to highlight again, that within each team, a great inter-individual ability to perform repeated intense exercise can be observed throughout the season, with some players improving, others decreasing and/or maintaining their performance (*Bangsbo, Iaia & Krustrup, 2008*).

Interestingly, *Boullosa et al. (2013)* did not report substantial changes in YYIR1 from PPS to BCP. It should be observed that in this study, players started the season with a high

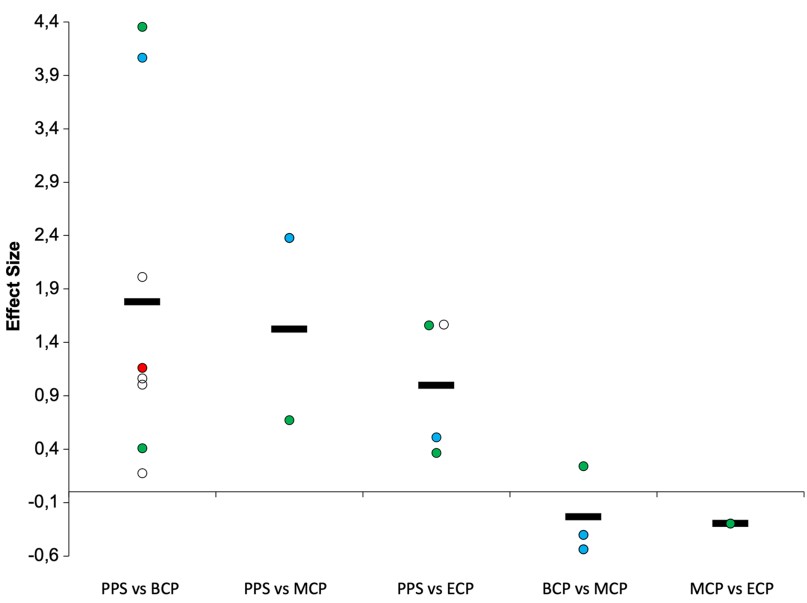

**Figure 12 Seasonal variations in intense intermittent endurance performance (weighted effect sizes).** PPS, prior preseason phase; BCP, beginning competition phase; MCP, middle competition phase; ECP, end of competition phase; white filled circles (YYIR1); blue filled circles (YYIR2); green filled circles (YYIE2); red filled circles (30–15 test); Dashed line represents average values.

YYIR1 performance (2,475-m), which may be related to the performance of an off-season program (5 weeks/21 sessions). This previous evidence, at least in part, highly indicate the benefits of performing a structured training program during the off-season (*Silva et al., 2016*). Moreover, it should be noted that despite no significant improvements in YYIR1, the authors reported important changes in certain indices of cardiac autonomic adaptations (*e.g.*, short heart-rate recovery) after this period of intensified training.

### Repeated sprint ability

*Impellizzeri et al. (2008)* observed that elite players improved different parameters in RSSA test performance throughout the season. Namely, the mean time of the sprints (RSSA$_{mean}$) improved to a moderate extent from PPS to BCP ($\Delta$ = 2.2%, ES = 1.14), MCP ($\Delta$ = 1.4%, ES = 0.74) and ECP ($\Delta$ = 1.6%, ES = 0.29). The fatigue index improved to a small magnitude from PPS to BCP ($\Delta$ = 20.4%, ES = 0.56), and in a moderate extent from PPS to MCP ($\Delta$ = 22.2%, ES = 0.62) and ECP ($\Delta$ = 25.9%, ES = 0.71). The lower fatigability during repeated sprints performed during MCP and ECP *vs* PPS as also been verified when monitoring U20 elite players using the Bangsbo sprint test (*Jorge, Garrafoli & Cal Abad, 2020*). Nevertheless, a small deterioration of the RSSA$_{mean}$ occurred from the BCP to MCP ($\Delta$ = 0.84%, ES = 0.41) with trivial changes been observed from MCP to ECP and for the fatigue index within these specific in-season moments.

We intended to characterize the general ability of performing repeated intense exercise and with this purpose we combined results of different specific IE tests that are widely used in professional settings. We acknowledge the differences between protocols of each

individual test and that they might evaluate slightly different physical capacities (*Buchheit & Rabbani, 2014*). As example, YYIR1 leads to a maximal activation of the aerobic system, whereas YYIR2 determines an individual's ability to recover from repeated exercise with a high contribution from the anaerobic system (*Bangsbo, Iaia & Krustrup, 2008*). Nevertheless, their sensitivity to training is almost certainly similar (30-15 *vs* YYIR1) (*Buchheit & Rabbani, 2014*) and given the very large correlations between tests (YYIR1 *vs* YYIR2) practitioners have been advised to consider using only one of the Yo-Yo tests and a RSA test in a general soccer-specific field test protocol (*Ingebrigtsen et al., 2012*; *Ingebrigtsen et al., 2013a*).

## Sub-maximal intermittent field exercise

It has been observed that soccer players $\%HR_{max}$ at the 6-min point of the YYIR1 decreased from the PPS to the middle of the pre-season, BCP and ECP (*Krustrup et al., 2003*). *Rago et al. (2020)* when applying the same protocol during the in-season period (four assessment moments from MCP to ECP) observed a continuous moderate improvement in heart rate measurements towards ECP. Moreover, others observed that even though professional players may show a decline in $VO_{2max}$ from the preparation period to the end of the season, their heart rate responses during the sub-maximal version of the YYIE2 were not altered during five time-points of a soccer season (from 14 days pre-season to ECP) (*Heisterberg et al., 2012*).

## Game-related physical parameters

Match analysis is a widely used instrument in professional soccer to study technical, tactical, and physical performances of players (*Abt & Lovell, 2009*). These instruments allow careful analysis of player match performance, dependent of a large number of factors (*e.g.*, training status, field position, age) and allows for the investigation of seasonal changes in game-related physical performance (*Helgerud et al., 2001*; *Impellizzeri et al., 2006*; *Morgans et al., 2014*; *Padron-Cabo et al., 2018*; *Rampinini et al., 2007b*; *Silva et al., 2013b*) and study evolutionary trends over consecutive seasons (*Akyildiz et al., 2022*; *Barnes et al., 2014*; *Bradley et al., 2016*; *Bush et al., 2014*; *Pons et al., 2021*; *Vigne et al., 2012*).

### Seasonal variations

Seasonal alterations in distance covered in different speed zones during the game is presented in Figs. 13–15 (*Link & de Lorenzo, 2016*; *Morgans et al., 2014*; *Padron-Cabo et al., 2018*; *Rampinini et al., 2007b*; *Silva et al., 2013b*). There are by average trivial changes in the TD (Fig. 14) from BCP to MCP ($\Delta = -1.09\%$, ES = 0.03, ranging from 9,150–10,513-m and 9,350–10,722-m, respectively) with a small increment between BCP and ECP ($\Delta = 1.63\%$, ES = 0.22, ranging from 9,600–10,921-m). A small increase in TD seems to occur from MCP to ECP ($\Delta = 1.5\%$, ES = 0.47). Interestingly, a clear variability exists, with both increments (*Mohr, Krustrup & Bangsbo, 2003*; *Rampinini et al., 2007b*; *Silva et al., 2013b*) and decrements (*Link & de Lorenzo, 2016*; *Mohr, Krustrup & Bangsbo, 2003*; *Padron-Cabo et al., 2018*) between these time-points. Importantly, within the season the variation ranged from −2.4% to 5.9% that are below for the reference value (10–15%)

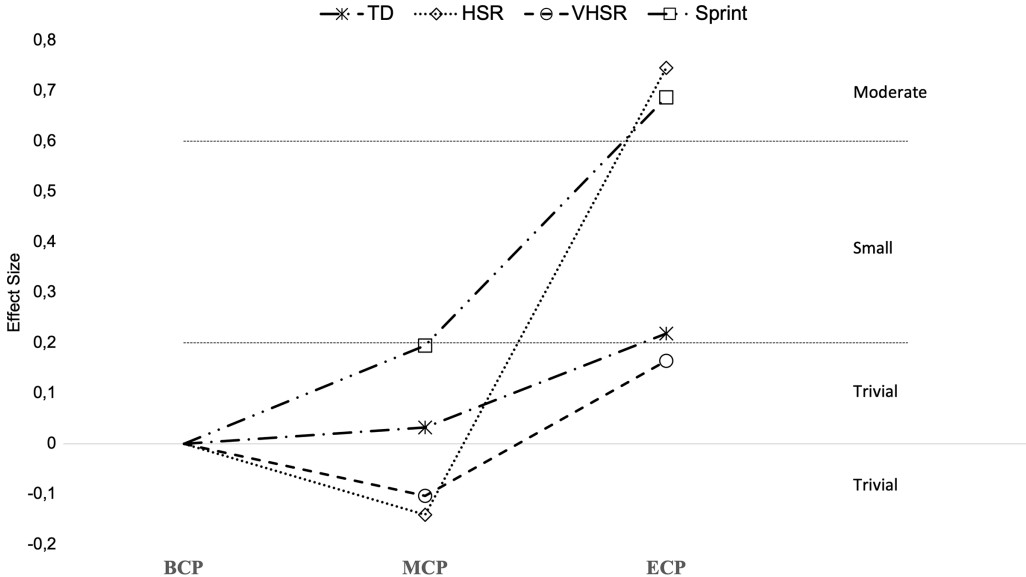

**Figure 13 Seasonal variations in game-related physical parameters (average weighted effect sizes).**
TD, total distance covered; HSR, high-speed running distance; VHSR, very-high-speed running distance; Sprint, sprint distance. BCP, beginning competition phase; MCP, middle competition phase; ECP, end of competition phase.

(*Oliva-Lozano et al., 2021a*) that can establish a practical significance considering the high match-to match variability (*Gregson et al., 2010*; *Oliva-Lozano et al., 2021a*).

The distance covered in HSR (~>14.4–15 km/h; Figs. 13 and 15) during the match has been proposed to be of great importance for performance in elite soccer because clearly distinguishes players of different standards (*Mohr, Krustrup & Bangsbo, 2003*; *Saeterbakken et al., 2019*). However, these observations of HSR proficiency being associated with player standards have not been unanimously confirmed (*Di Salvo et al., 2012*). The amount of HSR by average decreases with a trivial magnitude from BCP to MCP (Δ = 2.1%, ES = −0.14, ranging from 1,350–2,450-m and 1,270–2,544-m, respectively) and increase to a moderate extent to ECP (Δ = 22.5%, ES = 0.75, ranging from 1,900–2,738-m) and from MCP to ECP (Δ = 25.9%, ES = 0.92). Interestingly all the studies, although reporting different magnitudes (small to large), observed substantial alterations between these two time-points (*Mohr, Krustrup & Bangsbo, 2003*; *Rampinini et al., 2007b*; *Silva et al., 2013b*). Moreover, the amount of HSR performed in the last fifteen-minute period of each half, indicative of the ability to maintain performance during the game (*Krustrup et al., 2005*), was reported to be higher towards the ECP (*Silva et al., 2013b*). Additionally, in the ECP, a greater distance in HSR was covered in the peak and in the lowest fifteen-minute periods of the match than in the corresponding fifteen-minute periods at other season time points (*Silva et al., 2013b*). Furthermore, *Silva et al. (2013b)* observed that professional players were more engaged in high-intensity activities and had higher peak 5-min periods of HSR during the matches towards the last quarter of the season (ECP).
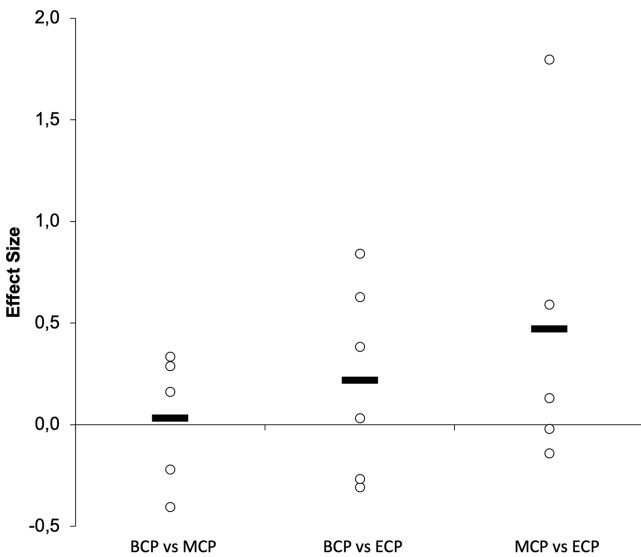

**Figure 14 Seasonal variations in total distance covered (weighted effect sizes).** BCP, beginning competition phase; MCP, middle competition phase; ECP, end of competition phase; Dashed line represents average values.

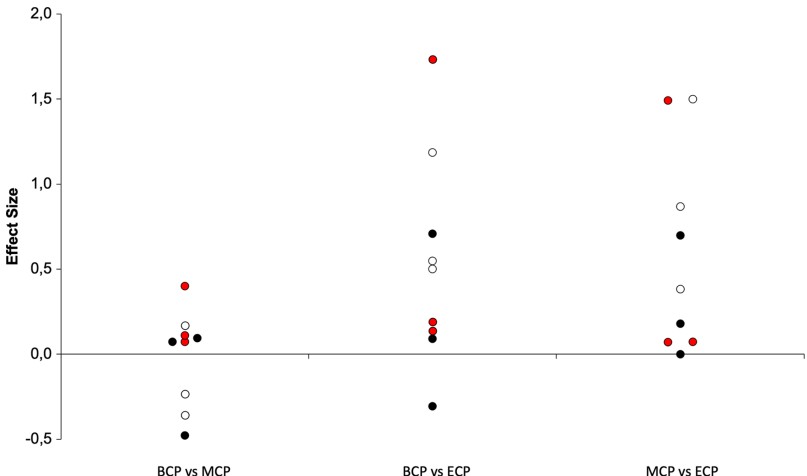

**Figure 15 Seasonal variations in high-intensity speed zones (weighted effect sizes).** BCP, beginning competition phase; MCP, middle competition phase; ECP, end of competition phase; white filled circles (HSR); black filled circles (VHSR); red filled circles (Sprint).

Very-high speed running (>19.8–21 km/h; VHSR; Fig. 15) is stable from the BCP to MCP ($\Delta$ = 3.8%, ES = 0.10, ranging from 465–916-m and 485–829-m, respectively) and ECP ($\Delta$ = 4.5%, ES = 0.16, ranging from 481–977-m). Small increments in VHSR may take place from MCP to ECP ($\Delta$ = 0.74%, ES = 0.29). Importantly, within the different season moments the variation ranged from −18% to 20% that are below for the reference value of 60–64% informing that a real change take place (*Gregson et al., 2010*; *Oliva-Lozano et al., 2021a*).

The sprint distance performed during the match (>24–30 km/h; Fig. 15) (*Morgans et al., 2014*; *Padron-Cabo et al., 2018*; *Silva et al., 2013a*) is by average stable from BCP to MCP (Δ = 4.5%, ES = 0.19, ranging from 98–201-m and 111–225-m, respectively) and increase with moderate and small magnitude from BCP to ECP (Δ = 11%, ES = 0.69, ranging from 192–234-m) and MCP to ECP (Δ = 6.1%, ES = 0.55), respectively. However, within the three studies analyzed, two (*Morgans et al., 2014*; *Padron-Cabo et al., 2018*) consistently reported trivial changes between these time-points. It should be again highlighted, that a large match-to-match variability in game-physical parameters of elite players may occur, suggesting that only large sample sizes may allow the clarification of systematic changes and that the "training stimulus" provided by the match is largely variable (*Gregson et al., 2010*; *Oliva-Lozano et al., 2021a*).

### Evolutionary trends

Evolutionary trends in match activity of professional players (*Akyildiz et al., 2022*; *Barnes et al., 2014*; *Bradley et al., 2016*; *Bush et al., 2014*; *Pons et al., 2021*; *Vigne et al., 2012*) has been analyzed in different contexts. *Vigne et al. (2012)* examined an Italian Serie-A team (2004–05 to 2006–07). The researchers observed significant progressive decreases in the distance covered per minute of play in low intensity running from the first to the second and third seasons. In addition, a significant decrease between the second and third seasons was also reported for moderate intensity running (*Vigne et al., 2012*); distance running and high intensity activities in Serie-A were similar in the three seasons. In spite no interaction between season and playing positions, and thus no signify alterations across all three seasons, were observed for the distinct field positions in the latter study (*Vigne et al., 2012*), others (*Bush et al., 2014*) performing a seven season longitudinal analysis (2006–07 to 2012–13) observed that the time dependent increase in physical demands (*e.g.*, increase in HSR) in the English premier league was extended to all players positions; full backs demonstrating the most pronounced increases. *Barnes et al. (2014)* carrying out the same design of *Bush et al. (2014)* observed an evolution of physical parameters in the English Premier League *e.g.*, across seven seasons high intensity running distance and actions increased by ~30% and ~50% and sprint distance and number increased by ~35% and ~85% respectively. Within this period (2006–13), *Bradley et al. (2016)* investigating the evolution of physical and technical performances in the same league with special reference to league ranking observed that physical and technical performances have evolved more in the 2[nd] Tier that included the teams from 5[th]–8[th] ranking than any other of the remaining three Tiers. According to the authors, this could indicate a narrowing of the performance gap between the top Tiers (*Bradley et al., 2016*).

*Pons et al. (2021)* examined evolutionary trends from 2015/2016 to 2018/2019 in the top two professional leagues of Spanish football. The authors observed a decrease in TD and an increase in the high-intensity distances and number of sprints performed, although a clearer trend was perceived in the top league. Additionally, VHSR and sprint distance increased during the second halves in both professional soccer leagues. Nevertheless, other authors did not observe an evolutionary trend in the Turkish league physical demands and independently of team's final rankings (2015–2018) (*Akyildiz et al., 2022*). All these studies

that independently investigate seasonal alterations in match activity and evolutionary trends in physical match performance, point out that players need more, to be ready to sustain activities involving a high metabolic and neuromuscular "cost". Interestingly an evolutionary trend in technical variables has been consistently reported (*Akyildiz et al., 2022*; *Barnes et al., 2014*), and that can be more evident in specific league tiers (*Bradley et al., 2016*).

Notwithstanding the intrinsic cultural characteristics associated with each league, differences between studies can be related, among other factors, to the (i) pre-defined thresholds of the different intensity categories of each analysis system (*e.g.*, high-intensity categories), (ii) discrepancies between systems in the accuracy of the determination of the distance covered at HSR (*Randers et al., 2010*), (iii) accuracy of the intensity of the pre-defined thresholds with individualized thresholds of physiological stress of the players (*Abt & Lovell, 2009*), and (iv) different game and situational conditions (*Paul, Bradley & Nassis, 2015*). In this regard, we need to highlight that several influencing factors may affect team and player performances at a behavioral level. Some of them are: (i) match status (*i.e.*, whether the team is winning, losing or drawing) (*Andrzejewski et al., 2018*; *Augusto et al., 2021*; *Bradley & Noakes, 2013*; *Oliva-Lozano et al., 2021b*), (ii) quality of opposition (*Lago-Penas et al., 2011*), (iii) match location (*i.e.*, playing at home or away) (*Augusto et al., 2021*; *Oliva-Lozano et al., 2021b*), (iv) fixture congestion (*Julian, Page & Harper, 2021*; *Lago-Penas et al., 2011*; *Oliva-Lozano et al., 2021b*), (v) environmental conditions (heat) and altitude (*Mohr et al., 2012*; *Nassis, 2013*), (vi) playing formation (*e.g.*, 1-4-2-3-1 *vs* 1-4-4-2) and style of play (*e.g.*, amount of ball possession) (*Arjol-Serrano et al., 2021*; *Bradley et al., 2013*), (vii) players availability (*Windt et al., 2018*), (viii) players physical fitness (*Altmann et al., 2018*; *Bradley et al., 2010*; *Konefal et al., 2019a*; *Krustrup et al., 2005*), (ix) distance traveled to play (*Augusto et al., 2021*) and (x) coach dismissal (*Augusto et al., 2021*; *Zart & Gullich, 2022*) and are all factors that may impact teams and players match output.

## Insights from training

High intensity training (HIT) comprises different modes of high-intensity exercise, namely high-intensity aerobic training (HIA), speed endurance training (SE) and repeated sprint ability training (RSA) (*Bishop, Girard & Mendez-Villanueva, 2011*; *Mohr et al., 2022*; *Spencer et al., 2005*). Generally, the common factors between modes are the high degree of physiological stress and the sharing of some similar physiological and functional training-induced adaptations imposed by the acute and chronic effects of the high-intensity bouts. HIT is a useful training method, providing a high training stimulus (*Bangsbo et al., 2009*; *Buchheit & Laursen, 2013a*; *Christensen et al., 2011*; *Iaia & Bangsbo, 2010*; *Iaia et al., 2009a*) on both the cardiopulmonary (*Buchheit & Laursen, 2013a*) and neuromuscular levels (*Buchheit & Laursen, 2013b*), thereby promoting physiological and performance adaptations that allow players to more successfully cope with the match and training demands (*Castagna et al., 2009*; *Girard, Mendez-Villanueva & Bishop, 2011*; *Gunnarsson et al., 2012*; *Helgerud et al., 2001*; *Iaia, Rampinini & Bangsbo, 2009*; *Impellizzeri et al., 2006*; *Ingebrigtsen et al., 2013b*; *Krustrup et al., 2005*; *Mohr, Krustrup & Bangsbo, 2003*; *Rampinini et al., 2007a*).

### Preseason

HIA, both in general (interval running; $HIA_{General}$) (*Helgerud et al., 2001*; *Impellizzeri et al., 2006*) and more specific modes (small-sided games and soccer-specific dribbling circuits; $HIA_{Specific}$) (*Impellizzeri et al., 2006*; *McMillan et al., 2005*), induces an improvement in several of the above analyzed physiological determinants (*e.g.*, $VO_{2max}$ and AT) and performance measures (*e.g.*, YO-YO tests; Ekblom's circuit test) in high level juniors and professional players. Moreover, the same as been observed when performing other forms of HIT (*Dupont, Akakpo & Berthoin, 2004*; *Wells et al., 2014*) or the concurrent performance of HIT with strength training (*Helgerud et al., 2011*; *McGawley & Andersson, 2013*; *Wong et al., 2010*). The latter seems to result in moderate (MAS and YYIR2) to very large (YYIR1) improvements endurance-related parameters (*Silva, 2019*). However, when adopting a concurrent training paradigm, soccer-related technical staff should implement an integrated approach when defining the exercise timing of the strength-based element of the session (see former Insights from training). Additionally, *Bogdanis et al. (2011)* observed improvements in physiological determinants ($VO_{2max}$) and endurance performance (YYIE2 and Hoff`s dribbling track test) by professional players after pre-season strength training independently of the target of adaptations (hypertrophy vs neural adaptations). Studies examining changes in the anaerobic running capacity are scarce. Nevertheless, improvements in 200- and 400-m running distances (*Sporis, Ruzic & Leko, 2008a*) and both the performance time and the ability to tolerate higher [La] during 300-y shuttle run test were reported to improve after pre-season $HIA_{Specific}$ (*Sporis, Ruzic & Leko, 2008a*, *2008b*). Furthermore, enhancements in a running-based anaerobic sprint test after 6-weeks pre-season $HIA_{Specific}$ of professional players have also been described (*Ostojic et al., 2009*). There are reports that pre-season $HIA_{Specific}$ improves high-level junior players performances of other forms of in-line running exercises (*e.g.*, 800, 1,200 and 2,400-m) with an important aerobic contribution (*Sporis, Ruzic & Leko, 2008a*, *2008b*).

### In-season

As was already mentioned, most longitudinal studies (*Aziz, Tan & Teh, 2005*; *Casajus, 2001*; *Metaxas et al., 2006*; *Mohr, Krustrup & Bangsbo, 2002*; *Silvestre et al., 2006*) and studies analyzing adaptations from specific training methodologies (*Impellizzeri et al., 2006*) did not detect significant further improvements after the initial increase in $VO_{2max}$ found after the pre-season phase. In fact, extending the preseason HIA of high-level junior players trough the initial weeks of in-season (7–8 weeks) did not produce any further substantial increase in the mean $VO_{2max}$ (2006). However, *Dupont, Akakpo & Berthoin (2004)* observed that professional players performing 2 weekly sessions of HIT for 10-weeks during the in-season period substantial increased MAS (~9%). Furthermore, *Jensen et al. (2009)* observed that U-20 elite players performing just one session of $HIA_{Specific}$ (30-min session per week) during the last 12-weeks of the competitive season, rather than the 2–3 weekly sessions traditionally applied in the other studies (*Dupont, Akakpo & Berthoin, 2004*; *Helgerud et al., 2001*; *Helgerud et al., 2011*; *Iaia, Rampinini & Bangsbo, 2009*; *Impellizzeri et al., 2006*; *Lopez-Segovia, Palao Andres & Gonzalez-Badillo, 2010*; *McMillan et al., 2005*; *Owen et al., 2012*; *Sporis, Ruzic & Leko, 2008a*, *2008b*; *Wong et al., 2010*),

substantial increase $VO_{2max}$ in addition to YYIR2 performance and improved fatigue time during RSA test. Additionally, an improvement in physiological measures (%$HR_{max}$ and blood lactate concentrations) during a sub-maximal version of the YYIR1 were observed (87.3% to 81.3% $HR_{max}$ and from 5 to 2.5 mmol/l, respectively) (*Jensen et al., 2009*). Particularly, it is likely that these different findings regarding in-season increments, namely in $VO_{2max}$ may be, among other factors, partially associated with the initial in-season $VO_{2max}$ of the distinct group of players (initial values of 52.8–55.7 ml/kg/min in *Ferrari Bravo et al. (2008)* and 59.7–61.4 in *Impellizzeri et al. (2006)*). In this regard, *Wells et al. (2014)* observed that the addition of 6-week speed endurance-based HIT to in-season training routines of professional players increased power, maximal speed, TE recorded during a maximal anaerobic sprint test, without improvement in certain physiological determinants of aerobic performance being examined (*e.g.*, $VO_{2max}$, MAS) (*Wells et al., 2014*). Within this period, a substantial increase in YYIR2 performance was also observed. Moreover, improvement in this field test was only associated with improvements in anaerobic capabilities (*Wells et al., 2014*). Furthermore, *Owen et al. (2012)* observed that $HIA_{Specific}$, conducted two times per week, during a four week in-season break, resulted in substantial improvements the total sprint time (1.8%) and the percentage of decrement score (~2.4 *vs* 1.5%) in an RSA test of elite professional players.

Among other factors, another important aspect is that most of the studies do not quantify the overall training load (*e.g.*, session and weekly training load) to which the players are exposed; that information may allow a better understanding of the different results between studies (*Martin et al., 2022*). Indeed, the time spent at high training intensities (pre-season) has been advocated as a powerful indicator for training monitoring; a positive association between physiological and performance improvements and the time spent training at high training intensities (*e.g.*, >90%$HR_{max}$) has been reported (*Casamichana et al., 2013*; *Castagna et al., 2011*; *Castagna et al., 2013*; *Manzi et al., 2013*). It has been recommended that professional players should spend at least a range of 7–8% of their total training volume during the pre-season in the high-intensity category (*Castagna et al., 2013*). Additionally, it seems that the weekly magnitude of the individualized training load (TRIMPi) of professional players should be higher than 500 AU, to substantially improve aerobic fitness and performance variables during the precompetitive season (*Manzi et al., 2013*). Moreover, an increase in weekly load by approximately 150-min in duration, 700 AU in sRPE, 12-km in total distance (TD), 2-km in HSR (>15 km/h; HSR) or 0.8-km min VHSR (>20 km/h) is required to increase the chances of obtaining a 0.5 mmol·l−1 improvement in the lactate accumulation during a 6 min constant speed running test (13.5 km/h) (*Martin et al., 2022*). Within this specific season period, an increase by 40-min in duration, 150 AU in sRPE, 3-km in TD, 1-km in HSR or 0.5-km in VHSR is required to increase the chances of obtaining a 0.5 mmol·l−1 improvement in the lactate accumulation during a high-intensity intermittent shuttle test (*Martin et al., 2022*).

Notwithstanding the previous studies, investigation of the effect of training programs in professional players is scarce, with more evidence during the preseason period. This is not surprising given that in professional/elite context due to the obvious limiting factors (*e.g.*,
physical demands of testing, limited time available, congested competition schedules) during in-season emphasis is given to prepare the strategy for next match and recovery from the stress of the last competitive match. Given these contextual limitations and that there is no common perspective or terminology to characterize the caliber and training status of an individual or cohort (*McKay et al., 2022*), interpret the existent training studies with a critical perspective is a crucial step for informed decision making.

In summary, during the preparation phase players "recover" cardiorespiratory capacity and the ability to perform and recover from high-intensity intermittent exercise. Improvements of moderate magnitude in velocity at fixed blood lactate concentrations ($V_{2-4mmol/l}$) and of large magnitude in $VO_{2max}$, maximal aerobic speed (MAS) and intense intermittent exercise performance (IE) are observed after preseason. During in-season, in MCP, are observed generally better scores when compared to PPS; improvements of moderate magnitude in $VO_{2max}$ and submaximal intensity exercise and large in MAS and IE. At ECP, increases are of moderate magnitude in all the examined outcomes. Although more scarcely investigated, from BCP to MCP, there are observed alterations of trivial magnitude in MAS (decrease) and $VO_{2max}$ (increase) and of small magnitude in IE (decrease) and sub-maximal exercise (increase). From the MCP to ECP, the different outcomes decrease with trivial ($V_{2-4mmol/l}$ and MAS) and small ($VO_{2max}$ and IE). Match performance may vary during the season. At the MCP the observed alterations are considered of trivial magnitude. However, it seems that at the ECP increments in TD (small), HSR (moderate), VHSR (small), and sprint (moderate) are of substantial magnitude compared the BCP. From the middle to the ECP, the observed increments are of small (TD, VHSR and sprint) and moderate magnitude (HSR). Although, the variability between studies is clear for TD, VHSR and sprint, all the studies observed substantial increments in HSR between the two previous time points. Different training methods or combination of methods may improve (pre-season) and assist in the maintenance or further improvement (in-season) of physiological determinants and endurance performance during the season.

## WHAT ARE THE CHALLENGES?

### Research

Research in soccer uncovers the complexity of interactions established between the different performance dimensions and the factors that are intrinsic to each player and team. However, the paucity of in-season data on specific anaerobic/neuromuscular qualities (*e.g.*, anaerobic power, relative force, rate force development, maximal speed) and physiological and endurance-related parameters (*e.g.*, RE, $VO_2$ kinetics, cardiac autonomic adaptations; short heart-rate recovery) that may be relevant in improving running capacity, should be investigated to allow for a better understanding of seasonal variations in physical fitness, more robustly, through the in-season phases. As example, overall systematic analyses of the data revealed better scores in multi-joint, power-based, dynamic efforts during in-season periods. In part, these observations may lead to the following proposals: (i) neuromuscular adaptations affecting SSC mechanisms (phase analysis) may occur throughout the in-season period; and (ii) a composite score of power-based efforts

may be more relevant for tracking the training status of professional players than a single measure, *per se*. Future research should also aim to understand seasonal changes in force capabilities during various velocities conditions and during specific motor tasks (jumping and sprinting) (*Morin, 2019*); efforts are already being developed in this direction (*Haugen, 2018*; *Jimenez-Reyes et al., 2022*). Moreover, studies aiming in improve the understanding of acute and chronic neuromuscular and endurance adaptations of professional players triggered by different in-season concurrent training modes (*e.g.*, two instead of one: build power and endurance at the same time) is necessary. Research examining the effect of match exposure throughout the season on the performance adaptation kinetics of professional players is warranted; match-playing time may influence adaptations of specific and non-specific endurance and neuromuscular parameters during the season (*Hader et al., 2019*; *Morgans, Di Michele & Drust, 2017*; *Silva et al., 2011*; *Sporis et al., 2011*). Furthermore, understand how the distinct internal and external load parameters (Level 1, 2 and 3 metrics) experienced by each individual player during the optimization of the distinct performance dimensions (*e.g.*, tactical) impact players fitness status will be key for optimize the full spectrum of the physical potential of the players (mechanical and metabolic). Studies characterizing the periodization of training loads (overall) during the pre-season and in-season periods of professional players are necessary. Moreover, considering the off-season detraining effects, a "reorganization" of the periodization during the transition period is necessary (*Silva et al., 2016*). In fact, these findings could lead one to question what the usefulness of such a loss of individual (collective) performance potential during off-season? We recently, made a call to action to understand how the prescription of off-season individualized training programs may influence seasonal performance (*Silva et al., 2016*). We highlighted that this period should be viewed as a 'window of opportunity' for players to recover and to 'rebuild' for the following season (*Silva et al., 2016*). 'Rebuild' for a more efficient and consistent in-season performance.

The perceptible increase in HSR towards the end-of-season period can be influenced, at least in part, by an improvement of pacing strategies in some form by professional players. As such, the development and improvement of conscious and/or sub-conscious pacing strategies (*Carling & Bloomfield, 2010*; *Edwards & Noakes, 2009*; *Mugglestone et al., 2012*) that seems to take place during matches cannot be excluded; there are contradictions regarding the concept of team sport players pacing their effort throughout the game (*Aughey, 2010*). This fact seems consistent with the higher physical performance in games towards the end of the season (*Mohr, Krustrup & Bangsbo, 2003*; *Rampinini et al., 2007b*; *Silva et al., 2013b*) and in other football codes (*Aughey, 2011*), without improvements in the majority of physiological and functional parameters; evidence of increases in certain stress biomarkers have also been reported (*Handziski et al., 2006*; *Heisterberg et al., 2012*; *Kraemer et al., 2004*; *Meyer & Meister, 2011*; *Reinke et al., 2009*; *Silva et al., 2014*; *Suda et al., 2012*). However, the well know context of the final stage of the competitive season (*e.g*, definition of team rank and contract renewal) as obvious impact in players "motivation" to perform. In these specific periods there is no space to "error", and most likely "Mind will prevail over Muscle" (*Marcora & Staiano, 2010*; *Pessiglione et al., 2007*).

As so, caution is needed when estimating players "readiness" from overall match activity profile. Research on these factors is necessary.

A better understanding of roles and tactics of team organization and an improvement in decision-making during season matches should be taken in account as central variables that may impact performance throughout the season (*Vigne et al., 2012*). Interestingly, data on longitudinal changes in match activity throughout the season seem to suggest an increased match efficiency (ranging from 2.6–6%) during the in-season period (efficiency = percentage of the total distance performed in high-intensity categories) (*Mohr, Krustrup & Bangsbo, 2003*; *Rampinini et al., 2007b*; *Silva et al., 2013b*). Another interesting factor is that high-level soccer players seem to exhibit superior anticipation capacity accompanied by more effective search behaviors and elaborative thought processes (*Casanova et al., 2013*). Nevertheless, the state of research regarding improvements in perceptual-cognitive processes in highly trained players and the influence of pacing and match activity remains very scarce. Curiously, elite players with long-term careers, parallelly to a annual gradual decrease in match-related physical output (0.56–1.8% by year) improve technical–tactical skills with increasing age (*Rey et al., 2022*). 'Integrated' approachs that contextualizes physical demands in relation to key tactical activities for each position and collectively for the team are warranted; understanding the physical performance in relation to the tactical roles (*Bradley & Ade, 2018*). In fact, is not the match running performance alone that is important for achieving success, but rather its relation to technical/tactical skills (*Hoppe et al., 2015*). Finally, the causative factors of the observed long-term changes (evolutionary trends) are not known and can be related, among others, with processes of players selection (*e.g.*, towards more "highly impulsive" players), improvements in facilities and equipment's (*e.g.*, grass conditions) and training-related processes (*e.g.*, better physical conditioning, training monitorization and players nutrition and recovery support). Research into the previous components is necessary.

## Training

Although one normally expects than within the season (from BCP to MCP or MCP to ECP) the consistent training of the physical, tactical, and technical dimensions of performance and as well the stimulus provided by competitive matches, could lead to a further optimization of players performance, more robustly concerning the start of competition phase (*e.g.*, BCP to MCP). However, within these periods there are by average observed changes of trivial magnitude. Specifically, substantial alterations where evident only for IE (decreased) and sub-maximal exercise performance (improved). From MCP to ECP all the examined parameters tend to decrease with a trivial magnitude and substantial negative alterations been observed for $VO_{2max}$ and IE. This undesirable dynamic in certain physiological determinants and endurance-related performance measures could be explained by the tight in-season schedule, with most of the time dedicate to recover from the previous match and prepare the strategy for the next opponent. In this regard, if a "window of opportunity" occurs (*e.g.*, player ban as result of a red card and players not selected for national team breaks) further in-season improvements in aerobic and

anaerobic qualities determinant for the running capacity and sub-maximal and maximal soccer-running performances than can be achieved through normal training routines may be obtained by incorporation of short duration HIT blocks (*Christensen et al., 2011*; *Wahl, Guldner & Mester, 2014*). As an example, although positive adaptations in running economy have mainly been reported and investigated during the pre-season, there are recent reports of increased running economy (75% of MAS) in players after performing 2 weeks of intense HIT executed just after the competitive season ended (*Christensen et al., 2011*); this result suggests that players still have significant physiological and performance adaptation potential to be explored. Nevertheless, caution is needed when extrapolating these findings for professional players as these experimental studies were performed by amateur and semi-professional players (*Christensen et al., 2011*; *Wahl, Guldner & Mester, 2014*). Nonetheless, it seems that special attention should be given to neuromuscular involvement during HIT (*Bogdanis et al., 2009*; *Bogdanis et al., 2011*) and to the concurrent effect of HIT (*McGawley & Andersson, 2013*), as it may be a determinant of the gains in running capacity during a short in-season intervention period. Nevertheless, being a very sensible process, these intervention periods require individualized management of the training/match load (*Silva & Rebelo, 2019*). In fact, within the same team a player may "underperform" as result of an over exposure while other player could be "underperforming" because of a detraining-related condition (*Silva & Rebelo, 2019*). Finally, the observations of a long-term persistent trend towards faster players and increased game speed (shorter and more "explosive" sprints and higher maximal running speeds) as well specific technical variables (*e.g.*, passing rates) should be reflected not only on players selection but also in the training organization (*e.g.*, physical conditioning). Regarding the latter, training should "feed" the players ability to perform maximal neuromuscular efforts and to repeat them over time; with the level of perceptual-cognitive demands varying according to each individual player needs.

## Monitoring

Notwithstanding some techniques applied in research settings provide valuable information (valid, reliable), its utilization in routine operations within the club setting is limited (*Silva & Rebelo, 2019*). The imposed physical demands (*e.g.*, maximal tests) and the invasive nature may explain at least in part the scarce applicability of several techniques in the real-world scenario (*Carling et al., 2018*). How motivated a player is for performing an end of season maximal testing session? This has obviously implications in the analyzed performance measures derived from testing sessions and match analysis, more influential through ECP assessments. Training may represent the perfect ecological setup to use as a 'lab' and shed some light as to the training status of the player (*Silva & Rebelo, 2019*). To this aim, a more action-oriented approach is needed; information derived from training sessions with tools that allow the simultaneous, instantaneous and non-invasive capture of multiple sources of information (*Carling et al., 2018*; *Morin et al., 2021*). As example, there are specific periods such as the warm-up and/or the main part of the training session (during gym or field sessions) that can be used to collect more precise information (neuromuscular and cardiorespiratory) regarding the players training status (*Silva &*

*Rebelo, 2019*). As example, *Morin et al. (2021)* recently investigate an *in-situ* approach to directly assess individual acceleration-speed profile. Moreover, standardized drills with planned (*e.g.*, sub-maximal running drill and or passing drills) (*Buchheit et al., 2013*) and unplanned (*e.g.*, small sided-games) external load (precise and imprecise behaviors, respectively) can be applied within those parts of training practice to gain insight on players training status (*Brink et al., 2012*; *Morin et al., 2021*; *Rago et al., 2017*; *Rago et al., 2018*; *Rowell et al., 2018b*). As example, given the clear disconnection between running economy assessment methodology and soccer-specific activity during training and matches, there is some evidence that soccer-specific work economy may somewhat improve during the season; the relevant gains may not be detectable by conventional treadmill testing (*Helgerud et al., 2001*; *McMillan et al., 2005*). Nevertheless, although this monitoring strategy is not applicable without the coach's approval, it builds an avenue for increased player "buy in" (*Silva & Rebelo, 2019*). A wide range of information can be collected during a standardized warm-up (*Buchheit et al., 2013*) that can inform on the training status of the athlete (*Halson, 2014*). As an example, the examination of physiological (HR) and perceptual (RPE) indicators of load in sub-maximal running exercise can provide valuable information on players cardiorespiratory fitness and fatigue level (*Halson, 2014*). Additionally, information on neuromuscular training status can be collected within this training stage and other periods (*e.g.*, standardized small-sided game) by means of GPS and accelerometer-derived metrics (*e.g.*, load per minute, load triaxial contributions) (*Cormack et al., 2013*; *Morin et al., 2021*; *Rowell et al., 2018b*). This latter monitoring strategy when applied during specific moments of the microcycle offers a great ecological and valid option for monitoring training status (*Rago et al., 2017*; *Rago et al., 2018*; *Rowell et al., 2018b*). In fact, this has been recently investigated in order to overcome the limitations (*e.g.*, time for testing, isolated tests) of assessing elite players when using more traditional moments and tools (*Rago et al., 2017*; *Rago et al., 2018*; *Rowell et al., 2018a*). Nevertheless, when using standardized small-sided games, well-known factors that affect players exercise intensity need to be regarded (*e.g.*, space, duration and team structure), but alsoteam constitution should be maintained stable (if possible the same players in each team) (*Silva & Rebelo, 2019*).

## LIMITATIONS

It is important to highlight some limitations inherent to this work. In this review, we aggregate teams from distinct soccer leagues (*e.g.*, European, and Asian). Although we included adults (>19 years) soccer players described has professional or elite player, given that there is no common perspective or terminology to characterize the caliber and training status of an individual or cohort, we need to consider that a considerable variation in training load and training history may exist between the included teams (*McKay et al., 2022*). Secondly, the time length between the different season moments may vary between studies. As example, in some studies the preparation period could last four weeks and in others height weeks. Furthermore, season organization may diverge. In some studies players could had an extended mid-season break (*e.g.*, 2 weeks) due to the winter environmental conditions (*e.g.*, Romanian League, and German Bundesliga), while in

others just a short number of days for Christmas festivities (*e.g.*, Portuguese, or Spanish leagues) or even mid-season being one of the most congested periods of the competition (*e.g.*, English premier league).

## CONCLUSION

Both short- and long-term detraining during the off-season period seem to have negative effects on body composition with alterations of small magnitude in body mass, body fat and decrements of moderate magnitude in lean body mass. The transition period also results in deteriorations of small to moderate magnitude in jump ability (Non-CMJ and $CMJ_{Based}$), linear (acceleration and maximal velocity phase) and multidirectional speed. Furthermore, a large magnitude in physiological determinants and endurance performance measures (large for $VO_{2max}$ and time to exhaustion and moderate and very large for intense intermittent exercise) have also been reported. These detraining effects may influence how players prepare during the pre-season and in-season and, in a certain way, affect their performance levels, especially in the first matches of the competitive season (*Kraemer et al., 2004*).

During the preparation phase players "recover" competitive capacity. The different investigations suggest that no unique and specific pattern of variation in body composition profile occur during the pre-season and in-season periods. Nevertheless, the general picture suggests that professional players may maintain their BM after the start of the training period through improvements of small magnitude in LBM and BF and with no substantial alterations within the in-season moments. These biometric alterations signify that chronic exposure of professional players to training and competition results in improved muscular and adiposity profiles and therefore a better overall body composition. Neuromuscular adaptations have been observed throughout absolute and relative measures of force production (1RM and relative force) as well as through jump, sprint, and COD tests. Specifically, by average, improvements of small magnitude in non-CMJ and $CMJ_{Based}$ jumps, and the acceleration and maximal velocity phase of the sprint are observed when preparing to competition phase. In the middle of the competition period, small ($CMJ_{Based}$ and $ACC_{Phase}$), and moderate (non-CMJ and $MV_{Phase}$) improvements were observed, compared to the start of the preseason phase. However, alterations towards the end of season (ECP) seem to be force-velocity dependent; $CMJ_{Based}$ and maximal speed improve to a small extent with non-CMJ and sprint acceleration phase revealing moderate performance increments compared to PPS. A general analysis suggest that trivial alterations occur withing the in-season (BCP to MCP and MCP to ECP) in these performance parameters. However, these is the result of the variability observed between studies; more evident when monitoring the CMJ performance.

Improvements of moderate magnitude in the velocity at fixed blood lactate concentrations ($V_{2-4mmol/l}$) and of large magnitude in $VO_{2max}$, MAS and IE are by average observed after preseason. During in-season, in the MCP, are observed generally better scores when compared to the PPS; by average, improvements of moderate magnitude in $VO_{2max}$ and submaximal intensity exercise and large in MAS and IE. At the ECP, the increases in the abovementioned parameters are of moderate magnitude in all the

examined outcomes. Although more scarcely investigated, from BCP to MCP, there are observed by average alterations of trivial magnitude in MAS (decrease) and $VO_{2max}$ (increase) and changes of small magnitude in IE (decrease) and sub-maximal intermittent exercise (increase). From the MCP to ECP, the different outcomes decrease by average with trivial ($V_{2-4mmol/l}$ and MAS) and small magnitudes ($VO_{2max}$ and IE). Match performance may vary during the season. At the MCP the observed alterations are by average considered of trivial magnitude. However, it seems that at the ECP, increments in TD (small), HSR (moderate), VHSR (small) and sprint (moderate) speed zones are of substantial magnitude compared the BCP. From the middle to the ECP, the observed increments are by average of small (total distance, VHSR and sprint) and moderate magnitude (HSR). Although, the variability between studies is clear for TD, VHSR and sprint, all the studies observed substantial increments in HSR between the two previous time points. Finally, studies examining evolutionary trends by means of exercise and competition performance measures suggests of a heightened importance of neuromuscular factors in soccer.

In conclusion, although an extraordinary growth in the number of scientific investigations concerning soccer has been observed in the $3^{rd}$ millennium, there is still much to elucidate regarding the complexity of interactions established between the different performance dimensions and the factors that are intrinsic to each player and team. Notwithstanding the fundamental role of the most-up-to date evidence-based training practices and monitoring tools for assure an efficient a proficient training process, high-level teams' success, and players excellence achievement, will be always closely dependent of the specificity of the training stimulus provided (*e.g.*, nature of the content) and sensibility of the technical staff (*e.g.*, mastery of coach managing players match/training load) on driving the training process.

### Funding
The authors received no funding for this work.

### Competing Interests
The authors declare that they have no competing interests.

### Author Contributions
- Joao Renato Silva conceived and designed the experiments, performed the experiments, analyzed the data, prepared figures and/or tables, authored or reviewed drafts of the article, and approved the final draft.

### Data Availability
   This is a literature review without raw data.

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
