# Peer review of "The soccer season: performance variations and evolutionary trends"

_PeerJ, doi:10.7717/peerj.14082_

## Round 0.1 · original submission · Minor Revisions

The article has merit and all the reviewers agree that only minor corrections should be taken into account.

Reviewers 1 & 3 requested that you cite specific references. You may add them if you believe they are especially relevant. However, I do not expect you to include these citations, and if you do not include them, this will not influence my decision.

·

Basic reporting

Clear, unambiguous, professional English
language used throughout.
Propably yes

Intro & background to show context.
Literature well referenced & relevant.
Yes

Structure conforms to PeerJ standards,
discipline norm, or improved for clarity.
Yes

Is the review of broad and cross-disciplinary
interest and within the scope of the journal?
The reference section should be supplemented with the indicated publications and entered into the text

Has the field been reviewed recently? If so,
is there a good reason for this review
(different point of view, accessible to a
different audience, etc.)?
Yes

Does the Introduction adequately introduce
the subject and make it clear who the
audience is/what the motivation is?
Yes

Experimental design

Article content is within the Aims and Scope of the journal.
Yes

Rigorous investigation performed to a high technical & ethical standard.
Yes

Methods described with sufficient detail & information to replicate.
Yes

Is the Survey Methodology consistent with a comprehensive, unbiased coverage of the subject? If not, what is missing?
I belive so

Are sources adequately cited? Quoted or paraphrased as appropriate?
Yes

Is the review organized logically into coherent paragraphs/subsections?
Yes

Validity of the findings

Impact and novelty not assessed. Meaningful replication encouraged where rationale & benefit to literature is clearly stated.
Submitted for review the literature review has great potential for citation

Conclusions are well stated, linked to original research question & limited to supporting results.
Yes

Is there a well developed and supported argument that meets the goals set out in the Introduction?
Yes

Does the Conclusion identify unresolved questions / gaps / future directions?
Vestigially described

Additional comments

Review for PeerJ

General comments:
The author gives citations from before 2000 or from the beginning of the 2000s, if this is a classic, a very recognizable article, leave it, but there is also a lot that can be removed - I strongly recommend that the old works be dried, because the game has changed a lot since then. The more recent the publications are, the better the quality of this manuscript will be
The rationale for undertaking the research is well-written
Punctuation errors were found in several places. Please review sentence by sentence again.
At the end of the manuscript, it should focus more on future research and briefly highlight the practical application
Overall, I think the article is good and should be published with minor changes


Specific comments:
Abstract
does the author provide too many results in the abstract?
please delete the least important
at the end of the abstract, there is no conclusion, a practical application
line 105 - i suggest to add https://www.ncbi.nlm.nih.gov/pmc/articles/PMC6458575/
line 115 - i suggest to add https://pubmed.ncbi.nlm.nih.gov/31185585/
line 124 - i suggest to add https://pubmed.ncbi.nlm.nih.gov/31185585/
line 240 - i suggest to add https://pubmed.ncbi.nlm.nih.gov/28488830/
I believe that this article should also be inserted into the text https://pubmed.ncbi.nlm.nih.gov/35539014/, it will strengthen the quality of the literature review
presentation of tables is poor. Too large spacing between columns, too small font, please correct

Reviewer 2 ·

Basic reporting

Having read the article " The soccer season: variations and evolutionary trends in physical fitness", I believe that the clarity and flow in some parts still need improvements. Hence, I cannot recommending it for publishing, according the following reasons:

The introduction section sintethyses the current state of the art.

Experimental design

Why the authors did not follow a specific guideline (e.g., PRISMA) to write thus review?

Why did you search in pubmed and medline? Medline is one of the databases inserted in pubmed….

How is it possible to perfume the search in 2013 if we are in 2022? For sure, this is a mistake.

Please update the search strategy do not repeating a set of word. Please combine the words with the Boolean operators and insert String Characters.

The results, discussion and conclusions are presented in a detailed way. Well done!

Validity of the findings

No comments

Additional comments

I suggest the introduction of a paragraph with the limitations of this study.

Reviewer 3 ·

Basic reporting

I read the entire manuscript with interest. It is very extensive and covers many topics. Although the title only contains physical fitness, the text also includes tactical and technical threads as well as topics related to decision-making by players.
Due to this multitude of information and a very large number of manuscript pages, important conclusions regarding each chapter are omitted during a long reading. Therefore, I suggest that you make a summary after each large chapter. In a few points, write the main conclusions and the main practical tips for the trainers.
You can see that the manuscript is written with coaching practice in mind. The title of the manuscript includes the phrase: "The soccer season." Please consider whether the main chapters in the manuscript should be the next parts of the season: eg the preparation period, the starting period, etc. Such an arrangement would be very useful from a coaching point of view.
In addition, it is worth considering adding a paragraph showing the interrelationships of the parameters described in the manuscript. Show work with modeling methods, because the tested parameters do not function separately and their influence power is important, e.g .:
Konefał M, Chmura P, Tessitore A, Melcer T, Kowalczuk E, Chmura J, Andrzejewski M. The Impact of Match Location and Players' Physical and Technical Activities on Winning in the German Bundesliga. Front Psychol. 2020 Jul 23; 11: 1748. doi: 10.3389 / fpsyg.2020.01748. PMID: 32793071; PMCID: PMC7390904.
Konefał M, Chmura P, Kowalczuk E, Figueiredo AJ, Sarmento H, Rokita A, Chmura J, Andrzejewski M. Modeling of relationships between physical and technical activities and match outcome in elite German soccer players. J Sports Med Phys Fitness. 2019 May; 59 (5): 752-759. doi: 10.23736 / S0022-4707.18.08506-7. Epub 2018 Jun 7. PMID: 29877676.
In the introduction, information about situational factors is written, it is another problem which, although it was mentioned, is not elaborated on in the article. After all, it is these factors that greatly modify the physical fitness of players. For example, the result of the match, the quality of the opposition, the climatic conditions.
In conclusion, I think the topic is very interesting and the work was written conscientiously with a large amount of literature. The methods were used correctly. If the editorial requirements of PeerJ allow the publication of such an extensive work, I recommend its publication.
Moreover::
- I am not a native speaker and I cannot judge the correctness of the English language,
- the literature is well used at work and describes the current state of knowledge well,
- it is an interesting project, which may be of particular interest to practitioners.

Experimental design

- In my opinion, the content of the article falls within the scope of the journal's goals,
- the article is written according to technical and ethical standards,
- the methods are properly described,
- sources are adequately cited,
- the review is logically organized, but with such an extensive text, maybe a table of contents could be useful.

Validity of the findings

- The findings are important both practically and cognitively. It is a good compendium of current knowledge in this field of research,
- the conclusions are well-formed,
- the goals are well argued,
- the article contains conclusions indicating further directions of research.

---

## Round 0.2 · accepted · Accept

The authors addressed the reviewer's comments and the quality and merit of the manuscript are apparent.

Reviewer 2 ·

Basic reporting

I would like to congratulate the author by this revised version of the paper.

Experimental design

I would like to congratulate the author by this revised version of the paper.

Validity of the findings

I would like to congratulate the author by this revised version of the paper.

Additional comments

I would like to congratulate the author by this revised version of the paper.